# A potent human neutralizing antibody Fc-dependently reduces established HBV infections

Dan Li[1,2], Wenhui He[2], Ximing Liu[2,3], Sanduo Zheng[2], Yonghe Qi[2], Huiyu Li[2], Fengfeng Mao[2,4], Juan Liu[2], Yinyan Sun[2], Lijing Pan[1,2], Kaixin Du[2,4], Keqiong Ye[2], Wenhui Li[2]*, Jianhua Sui[2]*

[1]Peking University-Tsinghua University-National Institute of Biological Sciences Joint Graduate Program, School of Life Sciences, Tsinghua University, Beijing, China; [2]National Institute of Biological Sciences, Beijing, China; [3]PTN Joint Graduate Program, College of Life Sciences, Peking University, Beijing, China; [4]Graduate Program in College of Life Sciences, Beijing Normal University, Beijing, China

**Abstract** Hepatitis B virus (HBV) infection is a major global health problem. Currently-available therapies are ineffective in curing chronic HBV infection. HBV and its satellite hepatitis D virus (HDV) infect hepatocytes via binding of the preS1 domain of its large envelope protein to sodium taurocholate cotransporting polypeptide (NTCP). Here, we developed novel human monoclonal antibodies that block the engagement of preS1 with NTCP and neutralize HBV and HDV with high potency. One antibody, 2H5-A14, functions at picomolar level and exhibited neutralization-activity-mediated prophylactic effects. It also acts therapeutically by eliciting antibody-Fc-dependent immunological effector functions that impose durable suppression of viral infection in HBV-infected mice, resulting in reductions in the levels of the small envelope antigen and viral DNA, with no emergence of escape mutants. Our results illustrate a novel antibody-Fc-dependent approach for HBV treatment and suggest 2H5-A14 as a novel clinical candidate for HBV prevention and treatment of chronic HBV infection.
DOI: https://doi.org/10.7554/eLife.26738.001

*For correspondence:
liwenhui@nibs.ac.cn (WL);
suijianhua@nibs.ac.cn (JS)

## Introduction

Human hepatitis B virus (HBV) infects more people than does Hepatitis C virus (HCV) and Human Immunodeficiency Virus (HIV) combined (*World Health Organization, 2016*). Despite the fact that there is an effective prophylactic HBV vaccine, 240 million people are chronically infected. HBV carriers are at high risk of developing severe liver diseases ranging from chronic hepatic insufficiency to cirrhosis and hepatocellular carcinoma (HCC) (*World Health Organization, 2016*; *Schweitzer et al., 2015*). Current therapies are inadequate for treating chronic HBV infection. For example, the immune-modulatory interferon alpha therapies have a curative effect in a small portion of patients; nucleos(t)ide analog (NUC) therapies can suppress viral replication but hardly achieve sustained virological control after treatment withdrawal and are non-curative (*Lok et al., 2016*). An additional problem involves co-infection with Hepatitis D virus (HDV), which is a satellite virus that propagates only in the presence of HBV. Approximately 15 million of HBV carriers are co-infected with HDV, which accelerates disease progression and exacerbates disease severity. No effective therapies are currently available for HBV patients co-infected with HDV, and there are no anti-viral therapies that specifically target HDV (*Rizzetto, 2015*; *Hughes et al., 2011*; *Thomas et al., 2015*).

HBV is a small enveloped DNA virus with a relaxed circular genome. Upon entering hepatocytes, HBV covalent closed circular DNA (cccDNA) is generated in the nucleus of infected cells, and this

cccDNA serves as both the intermediate for viral replication and as a viral persistence reservoir. It is believed that curing chronic hepatitis B (CHB) would require either elimination or permanently silencing of the cccDNA from hepatocytes and/or elimination of the infected hepatocytes carrying cccDNA (*Zeisel et al., 2015*; *Seeger and Mason, 2016*; *Liang et al., 2015*; *Block et al., 2013*; *Gane, 2017*).

Immunotherapies based on human monoclonal antibodies (mAb) have achieved remarkable clinical success in treating multiple cancers and various autoimmune and inflammatory diseases. More recent efforts have shown that chronic viral infectious diseases can be treated via mAb-based immunotherapy; studies have reported promising clinical outcomes using a mAb with broad and potent activity in neutralizing HIV (*Schoofs et al., 2016*; *Scheid et al., 2016*; *Caskey et al., 2015*). Neutralizing antibodies (nAbs), in addition to their capacity to specifically block viral entry via Fab (fragment of antigen binding) recognition of virus, have been found to exert a variety of immunological 'effector' functions, including clearance of circulatory viruses as well as by mediating cytotoxic killing or phagocytosis of infected cells (*DiLillo et al., 2014*; *Corti et al., 2011*; *Bournazos et al., 2014*; *Bruel et al., 2016*; *Lu et al., 2016*; *Hessell et al., 2007*), or even possibly triggering sustained host immune responses in vivo (*Schoofs et al., 2016*; *Pelegrin et al., 2015*). For example, just as with cancer cells, virus-infected cells can be eliminated by antibody dependent cell-mediated cytotoxicity (ADCC) and phagocytosis (ADCP) through interaction of the Fc (fragment crystallizable domain) of an antibody with its cognate Fcγ receptors (FcγRs) expressed on immune cells. Therefore, although the results from other immune-based therapies in treating HBV have so far been disappointing (*Zhang et al., 2015*), novel nAb-based immunotherapy may represent a new modality for curing chronic HBV infection.

Infection of HBV is mediated by HBV envelop proteins, which are the main target of neutralizing antibodies. No structural data are available for these multi-transmembrane proteins. The large (L) envelope protein of HBV has a preS1 domain at the N-terminal end of S domain and plays a pivotal role in HBV and HDV infections (*Le Seyec et al., 1999*; *Blanchet and Sureau, 2007*) (*Figure 1—figure supplement 1*). We discovered that the preS1 domain of the L protein specifically binds to the liver bile acid transporter sodium taurocholate cotransporting polypeptide (NTCP) in hepatocytes and thusly mediates viral entry of HBV and HDV (*Yan et al., 2012*; *Li, 2015*). Given this key functional role, the preS1 domain containing the NTCP-binding site is an attractive target for developing nAbs against HBV and HDV infection.

In the present study, we took advantage of a large non-immune phage display human antibody library and our human NTCP-enabled HBV cell culture system (*Yan et al., 2012*; *Sun et al., 2016*) to identify a panel of nAbs that specifically target the preS1 domain. One of the lead nAbs, 2H5, showed strong neutralizing ability against both HDV and HBV. We used crystallographic studies to identify the 2H5 epitope, and sequence analysis predicted that this epitope is highly conserved in eight of the ten HBV genotypes. Neutralization-oriented optimization of the lead nAb by antibody engineering resulted in a more potent anti-preS1 nAb, 2H5-A14, with picomolar neutralizing activity against the major epidemic genotypes of HBV (B, C, and D). Importantly, 2H5-A14 demonstrated ADCC activity through natural killer (NK) cells and through macrophages, and ADCP activity through macrophages in vitro. In animal studies, the prophylactic use of 2H5-A14 resulted in full protection of humanized mice from HDV and HBV infection. Moreover, 2H5-A14 greatly reduced HBV infection in a treatment mouse model. Studies with 2H5-A14 Fc mutant in the animals revealed that Fc-mediated immune effector functions are largely responsible for the therapeutic effect of 2H5-A14.

## Results

### Identification of a human nAb against the preS1 domain of the HBV L protein and structural analysis of the antigen-antibody complex

To identify nAbs that block viral infection, we used preS1 peptides derived from the preS1 domain of the HBV L protein (*Figure 1—figure supplement 1*) as targets to select against a large non-immune phage display antibody library. By employing HepG2-hNTCP cell-based in vitro HBV infection systems (*Sun et al., 2016*), we identified six nAbs that were effective in neutralizaling HBV, but each differed in its potency (*Figure 1A*). Epitope mapping by direct binding and competition ELISA assays showed that these nAbs recognized four unique epitopes, represented by #71, 2H5, m1Q

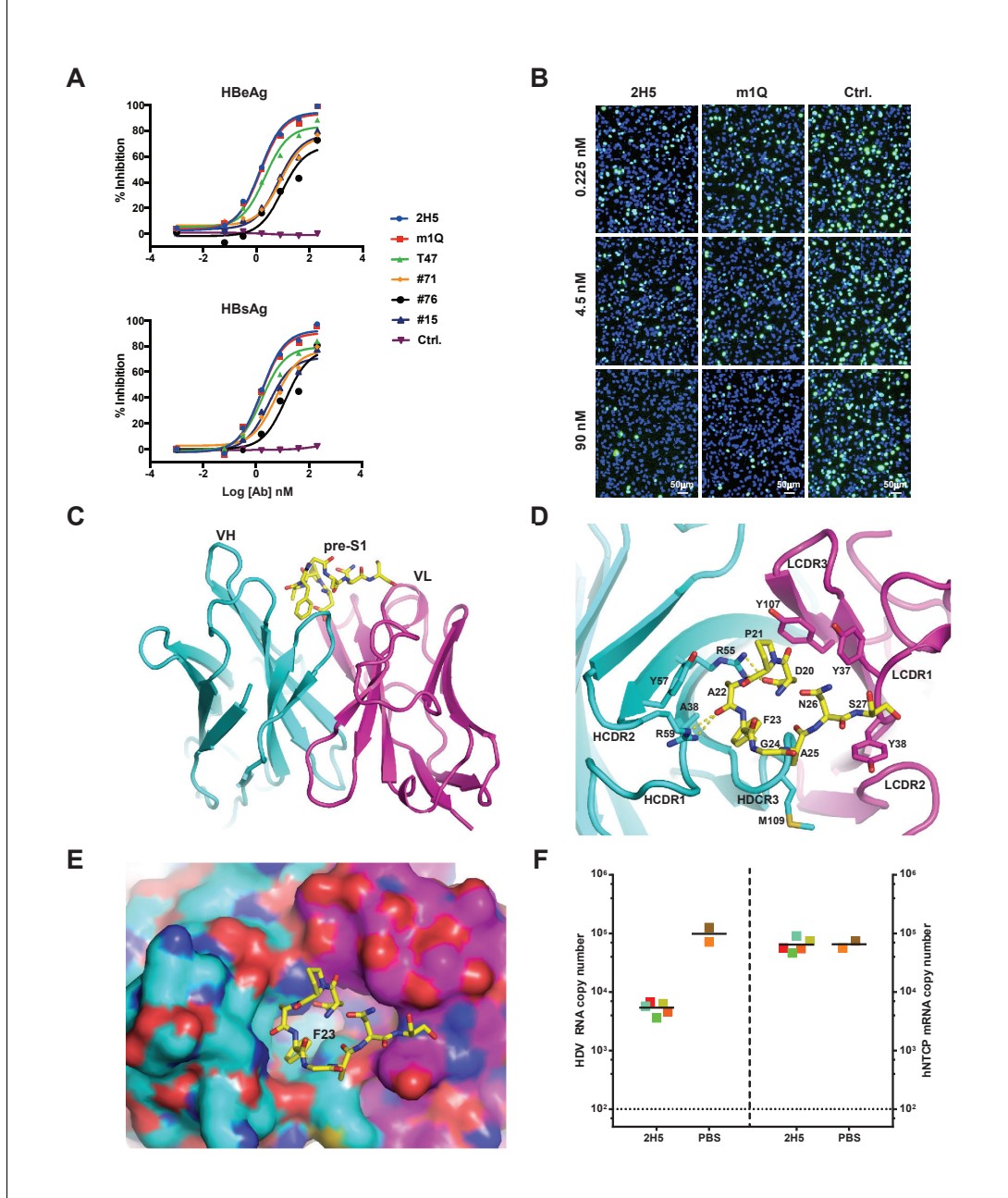

**Figure 1.** Identification, in vivo activity verification, and structural characterization of 2H5. (**A**) Neutralization of HBV (genotype D) infection in the HepG2-hNTCP stable cell line by anti-preS1 Abs. Cells were infected with 200 multiplicities of genome equivalents (mge) in the presence of the tested Abs. The secreted HBeAg and HBsAg levels in sample supernatants were measured at 7 days post infection (dpi). The HBV neutralization activity is here presented as the percentage of inhibition of secreted HBeAg and HBsAg, which was calculated by normalizing the 'infection only' reading to 0% inhibition. (**B**) Neutralization of HDV infection in HepG2-hNTCP cells by 2H5 and m1Q. The cells were infected with 500 mge of HDV in the presence of the test nAbs or matched isotype control Ab. At dpi 7, HDV delta antigens were stained with FITC-conjugated mAb, 4G5; the cell nuclei were stained with DAPI. (**C**) Crystal structure of the scFv of 2H5 in complex with the preS1 peptide. 2H5 is shown in ribbon and residues 20–27 of preS1 (genotype C) are shown in sticks. Carbon atoms are colored in cyan for VH of 2H5, magenta for VL of 2H5 and yellow for the preS1 peptide. Oxygen atoms are colored in red and nitrogen atoms are colored in blue. (**D**) Interaction between 2H5 and preS1. 2H5 is shown in ribbon and the preS1 epitope and selected CDR side chains are shown in sticks. This view is perpendicular to panel C. Dotted lines denote hydrogen bonds. (**E**) Surface view of the peptide binding pocket of 2H5. Same color-coding and orientation as panel D. (**F**) 2H5 protected *hNTCP-Tg* mice from HDV infection. Human *NTCP-Tg* homozygotes (C57BL/6 background) were IP administered 2H5 (20 mg/kg; n = 5) or PBS control (n = 2) at 8 days after birth. One hour later, each mouse was challenged with $1.47 \times 10^{10}$ genome equivalents (GE) of HDV. The mice were sacrificed at dpi 6. HDV total RNA titers (on the left Y-axis) and *hNTCP* transgene expression (on the right Y-axis) in liver tissues were measured by qPCR. The horizontal dotted line indicates the reliable

*Figure 1 continued on next page*

Figure 1 continued

detection limit. The copy numbers shown in the Y-axes are from 20 ng of total liver RNA for each sample. Each square represents one mouse; squares of the same color indicate data from the same mouse.

DOI: https://doi.org/10.7554/eLife.26738.002

The following source data and figure supplements are available for figure 1:

**Source data 1.** Data for *Figure 1F*.

DOI: https://doi.org/10.7554/eLife.26738.005

**Figure supplement 1.** HBV envelope proteins and amino acid sequence alignment of preS1 peptides.

DOI: https://doi.org/10.7554/eLife.26738.003

**Figure supplement 2.** Binding of the six anti-preS1 nAbs to preS1 peptides and characterization of their epitopes.

DOI: https://doi.org/10.7554/eLife.26738.004

and #15, respectively (*Figure 1—figure supplements 1* and *2A–C*). The neutralization potency of these nAbs correlated with the location of their epitopes in relation to the receptor binding site (RBS) in preS1. The epitopes of 2H5 and m1Q are more closer to the C-terminus of RBS, and showed more potent neutralization activity compared to the other two nAbs, including #71, with its epitope close to the N-terminus of RBS that only recognizes non-myristoylated preS1 peptides, and #15, with its epitope distant from the C-terminus of RBS, although all four of these nAbs had similar preS1 binding activity. 2H5 and m1Q were also able to neutralize HDV infection in HepG2-hNTCP cells (*Figure 1B*). As 2H5 was slightly more potent in neutralizing HBV and HDV than m1Q, we selected 2H5 for further characterization and optimization.

To better understand the structural basis of 2H5-preS1 binding, we solved the crystal structure of 2H5 (in scFv form (single-chain fragment of variable domain): VH-linker-VL) in complex with a preS1 peptide ('59C') corresponding to residues (−10 ~ 48) of pre-S1 (genotype C) (*Figure 1—figure supplement 1*). The 2H5 scFv and the 59C peptide were co-expressed in *E. Coli* and were then co-purified and co-crystallized. The complex structure was determined by molecular replacement and refined at 2.5 Å resolution to an R-factor of 0.23 and an R-free value of 0.28 (*Table 1*). In the complex structure, residues 20–27 of the 59C peptide showing well-ordered electron density were modeled. The 8-residue peptide adopts a U-shaped conformation and is situated in a pocket formed by five complementarity-determining-region (CDR) loops from both the variable heavy (VH) and light chains (VL) of 2H5 (*Figure 1C–E*). The interface between 2H5 scFv and the preS1 peptide buries a solvent accessible area of about 460.8 Å$^2$, accounting for 54.6% of the total peptide surface area. The heavy chain and light chain of 2H5 contribute 236.3 Å$^2$ and 224.5 Å$^2$ of the buried surface area respectively.

Five CDR loops (HCDR1-3, LCDR1, and LCDR3) are involved in direct binding to the 59C peptide. The 2H5-preS1 interaction interface includes contributions from all types of non-covalent interactions (*Figure 1D* and *Table 2*). All of the eight modeled residues except Gly$^{24}$ of preS1 interact directly with 2H5. On the buried surface of preS1, which is mainly involved in heavy chain interaction, the Asp$^{20}$ residue has an electrostatic interaction with Arg$^{55}$ in the HCDR2 loop. Additionally, Arg$^{59}$ of the HCDR2 loop forms a hydrogen bond with the main-chain carbonyl oxygen of Ala$^{22}$ from preS1. Phe$^{23}$ in preS1, which has a large hydrophobic benzyl group, inserts deep into a hydrophobic pocket that is formed by Ala$^{38}$, Arg$^{55}$, Tyr$^{57}$ and the backbone atoms of HCDR3. Phe$^{23}$ alone contributes to 23.5% of the buried surface area of the preS1 peptide. Ala$^{22}$ and Ala$^{25}$ in preS1 form weak hydrophobic interactions with, respectively, Tyr$^{57}$ of HCDR1 and Met$^{109}$ of HCDR3. On the exposed surface of preS1, which primarily contacts the light chain, Pro$^{21}$ forms hydrophobic interactions with Tyr$^{37}$ of LCDR1 and Tyr$^{107}$ of LCDR3. Additionally, Asn$^{26}$ and Ser$^{27}$ make several van der Waals contacts with both Tyr$^{37}$ and Tyr$^{38}$ of LCDR1.

We next verified the key binding residues deduced from the crystal structure of the complex. A wild-type synthesized 16-mer preS1 peptide could efficiently compete with the full-length preS1 peptide for binding to 2H5. However, three 16-mer variants of the preS1 peptide that had a key residue (Asp$^{20}$, Pro$^{21}$, or Phe$^{23}$) mutated to alanine failed to compete the full-length preS1 peptide for binding to 2H5 (*Figure 1—figure supplement 2D*). Further, alanine substitution of preS1 Leu$^{19}$, which is immediately adjacent to the 8-residue region of the 59C peptide, also reduced binding competition activity, indicating that this residue is involved in preS1-2H5 binding. Of note, sequence alignment showed that 2H5 epitope is highly conserved among HBV genotypes (*Figure 1—figure*

**Table 1.** Data collection and refinement statistics.
Values in parentheses are for the highest-resolution shell.

| | preS1-2H5 scFv |
|---|---|
| **Data collection** | |
| Space group | $P2_12_12$ |
| Cell dimensions | |
| a, b, c (Å) | 142.7, 55.7, 67.7 |
| $\alpha$, $\beta$, $\gamma$ (°) | 90, 90, 90 |
| Wavelength (Å) | 0.9793 |
| Resolution range (Å) | 20–2.5 (2.54–2.50) |
| Unique reflections | 18558(886) |
| Redundancy | 5.8 (5.9) |
| $<I>/<\sigma(I)>$ | 17.7 (6.2) |
| Completeness (%) | 96.0 (94.9) |
| $R_{merge}$ | 0.144 (0.345) |
| **Refinement** | |
| Resolution range (Å) | 20–2.5 (2.90–2.50) |
| No. reflections | 18265 |
| No. atoms | 3747 |
| Protein | 3519 |
| Water | 227 |
| Ion | 1 |
| $R_{work}$ | 0.231 (0.292) |
| $R_{free}$ | 0.283 (0.356) |
| Mean B factor (Å$^2$) | 14.5 |
| Rmsd bond length (Å) | 0.002 |
| Rmsd bond angles (°) | 0.555 |

DOI: https://doi.org/10.7554/eLife.26738.006

supplement 1), indicating that 2H5 can potentially provide broad protection against the vast majority of HBV genotypes. Although residue 24 in the 2H5 epitope is variable among different HBV genotypes (e.g., Gly$^{24}$ in genotypes A and C; Lys$^{24}$ or Arg$^{24}$ in all other genotypes, *Figure 1—figure supplement 1*), G24R mutation of the preS1 peptide had no effect on binding to 2H5 (*Figure 1—figure supplement 2E*), consistent with our observation from the complex structure that Gly$^{24}$ does not make direct contact with 2H5.

**Table 2.** Contact residues between 2H5 and PreS1 peptide.
Contact residues are here defined by interatomic distances of less than 5 Å. Residues involved in salt bridge and hydrogen bond interactions are bolded; the remaining residues engage in van der Waals or hydrophobic interactions.

| 2h5 scFv | HCDR1 | HCDR2 | | | HCDR3 | | | | CDR-L1 | | CDR-L3 |
|---|---|---|---|---|---|---|---|---|---|---|---|
| | **A38** | **R55** | **Y57** | **R59** | **G107** | **Q108** | **M109** | **G113** | **Y37** | **Y38** | **Y107** |
| PreS1 | F23 | **D20** P21 A22 F23 | A22 | **A22** F23 | F23 | F23 | F23 A25 | **D20** F23 | P21 N26 | S27 | **D20** P21 |

DOI: https://doi.org/10.7554/eLife.26738.007

## Generation of a more potent nAb, 2H5-A14

To evaluate the neutralization activity of 2H5 in vivo, we employed a de novo HDV infection mouse model that we recently established (*He et al., 2015*) in which the transgenic expression of human NTCP (*hNTCP*-Tg) in C57BL/6 mouse liver supports acute HDV infection mediated by HBV envelope proteins. Using this model, prophylactic administration of 2H5 IgG1 (1 hr before HDV viral challenge) resulted in a more than one log reduction of the viral titer in mouse livers as compared to untreated controls (*Figure 1F*). Nevertheless, 2H5 treatment was not able to completely block HDV infection in mouse livers, even at a 20 mg/kg dose. Thus, increasing the neutralization activity of 2H5 while preserving its conserved epitope were needed to yield a potent and broadly-neutralizing antibody with improved in vivo efficacy.

To this end, we employed various antibody engineering approaches including chain shuffling, structure-guided design, and CDR-focused mutagenesis. The VH chain shuffling approach was most fruitful in identifying improved anti-preS1 nAbs. VH chain shuffling by fixing the variable light (VL) chain of 2H5 and pairing it with a repertoire of naturally-occurring variable heavy (VH) genes ($\sim 1 \times 10^{10}$) led to the identification of 16 Abs with higher binding affinity. HBV and HDV neutralizing assays were then performed to rank the neutralizing ability of these 16 Abs relative to that of 2H5 (*Figure 2—figure supplement 1A*). The viral neutralization activities of the 11 out of the 16 Abs were then further confirmed in their human IgG1 form (*Figure 2—figure supplement 1B*). ELISA assays revealed that the four top (2H5-31, 2H5-32, 2H5-A14, and 2H5-A21) nAbs had about 100-fold improvement in binding activity relative to 2H5 (*Figure 2-figure supplement 2A*). Further neutralization assays revealed that, relative to 2H5, these four nAbs all had about 20-fold increases in neutralizing activity against HBV and about 100-fold increases against HDV, and their HBV neutralization $IC_{50}$ values were all lower than 50 pM under the conditions tested (*Figure 2-figure supplement 2B–C*). Subsequent differential scanning calorimetry (DSC) experiments with the four nAbs revealed good thermostability of 2H5-A14. It is noteworthy that the major amino acid sequence differences of the four nAbs and 2H5 are in their HCDR3 loops, particularly 2H5-A14 only differs from 2H5-31 and 2H5-A21 in its HCDR3 (*Figure 2—figure supplement 3*).

## 2H5-A14 is a potent and broadly-neutralizing nAb against HBV infection that acts by blocking the binding of preS1 with the HBV receptor NTCP

Competition ELISA assays showed that 2H5-A14 had almost the same peptide competition pattern as did 2H5, and mutation of G24R of preS1 did not affect 2H5-A14 binding activity, indicating that 2H5-A14 targets the same epitope as does 2H5 (*Figure 2—figure supplement 4*). This epitope is in close proximity to, but does not overlap, the receptor NTCP-binding site of preS1 (*Figure 1—figure supplement 1*). To test whether the neutralization activity of 2H5-A14 can be attributed to competition with preS1 for binding with NTCP, we used a stable HepG2-hNTCP cell clone that expresses a high level of hNTCP and supports the binding of a fluorescent-labeled preS1 peptide. An immunofluorescence staining assay showed that 2H5-A14 inhibited the binding of FITC-labeled preS1 to HepG2-hNTCP cells in a dose-dependent manner, whereas an isotype-matched control Ab did not (*Figure 2A*). This result indicates that the neutralization mechanism of 2H5-A14 likely involves blocking of viral entry by interfering with the binding of NTCP and preS1, probably owing to steric hindrance at or near the preS1 receptor binding site.

We next examined the potency and breadth of 2H5-A14 in neutralizing HBV and HDV. HBV has 10 known genotypes, all of which differ from each other by at least 8% at DNA sequence level (*Sunbul, 2014*). Genotypes B, C, and D represent the major epidemic genotypes, and 2H5-A14 potently neutralized all three of these genotypes (*Figure 2B*). We also compared the potency of 2H5-A14 to other agents known to block HBV entry, including commercial HBIG (Hepatitis B Immune Globulin), KR127 (a humanized preS1 mouse mAb [*Chi et al., 2007*]), and two prototype peptides of Myrcludex-B (*Bogomolov et al., 2016*) (an NTCP-binding preS1-derived synthetic myristoylated lipopeptide): m47-D (genotype D) and m47 (genotype C). Assays using a HepG2-hNTCP cell-based HBV infection system showed that 2H5-A14 had more than 1000-fold stronger neutralization activity than did HBIG against infection with recombinant HBV (*Figure 2C*) and HDV (*Figure 2D*), as well as against infection with six HBV primary isolates derived from HBV-infected patients (genotype C) (*Figure 2—figure supplement 5*). 2H5-A14 also had about 200-fold stronger neutralizing activity against

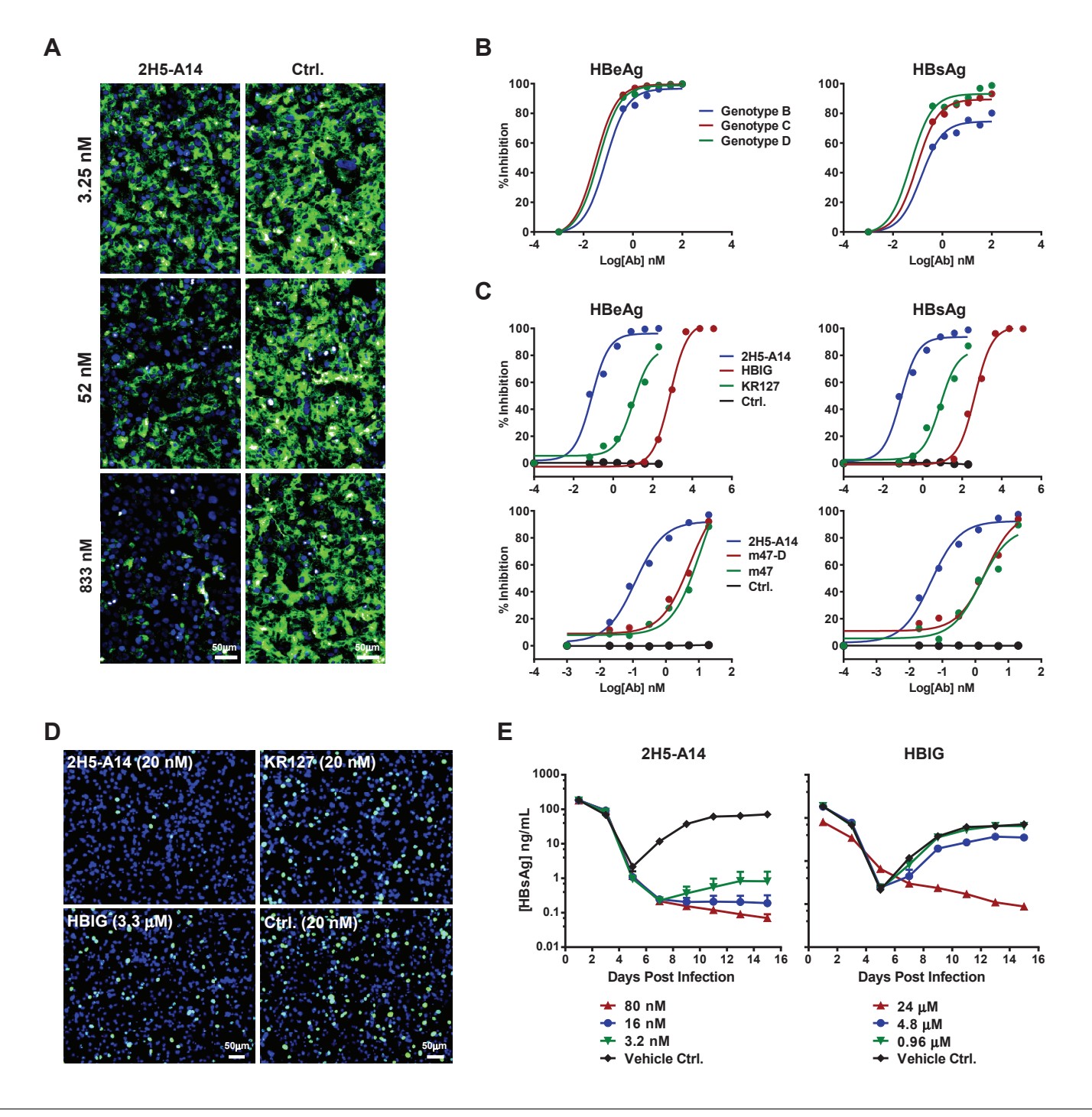

**Figure 2.** Blocking activity of 2H5-A14 for preS1 binding to hNTCP and the broad and potent neutralization activity of 2H5-A14 against HBV and HDV. (A) 2H5-A14 competed for the binding of preS1 to HepG2-hNTCP cells. 2H5-A14 or an isotype-matched control Ab at the indicated concentrations and 100 nM of FITC-labeled preS1 lipopeptide were mixed, and added to HepG2-hNTCP cells and were incubated for 15 mins, followed by extensive washing and staining of nuclei by DAPI. (B–E) The broad and potent neutralization activity of 2H5-A14 against HBV and HDV infection. 2H5-A14 neutralized HBV genotypes B, C, and D (B). In comparison to the prototype preS1 lipopeptides of Myrcludex-B, KR127 and HBIG in neutralizing HBV infection of HepG2-hNTCP cells, 2H5-A14 showed superior activity (C). 2H5-A14 also potently neutralized HDV infection of HepG2-hNTCP cells (D) and HBV infection of PHHs (E). The neutralization assays were performed similarly as in *Figure 1A–B*.

DOI: https://doi.org/10.7554/eLife.26738.008

*Figure 2 continued on next page*

*Figure 2 continued*

The following source data and figure supplements are available for figure 2:

**Source data 1.** Data for *Figure 2B, 2C and 2E*.

DOI: https://doi.org/10.7554/eLife.26738.014

**Figure supplement 1.** Neutralization of HBV and HDV infection by the 11 nAbs derived from 2H5-VH chain shuffling.

DOI: https://doi.org/10.7554/eLife.26738.009

**Figure supplement 2.** preS1 binding activity and neutralization activities of the top four nAbs obtained from 2H5-VH chain shuffling.

DOI: https://doi.org/10.7554/eLife.26738.010

**Figure supplement 3.** Amino acid sequence alignment of the VHs of nAbs.

DOI: https://doi.org/10.7554/eLife.26738.011

**Figure supplement 4.** Epitope mapping of 2H5-A14.

DOI: https://doi.org/10.7554/eLife.26738.012

**Figure supplement 5.** 2H5-A14 neutralization of patient serum derived HBV viruses, and sequence variation analysis of the 2H5-A14 epitope among HBV genotypes.

DOI: https://doi.org/10.7554/eLife.26738.013

HBV than did KR127, and 50-fold stronger anti-HBV activity than did either m47-D or m47 (*Figure 2C*). Consistently, in a primary human hepatocyte (PHHs) infection system, 2H5-A14 showed similar neutralization potency against a clinical viral isolate from an HBV-infected patient (genotype B) (*Figure 2E*).

We examined variations in the epitope-forming amino acid sequences among HBV genotypes to assess the breadth of the neutralization activity of 2H5-A14 against HBV. In addition to the aforementioned variability at position 24 of preS1, position 22 also exhibited heterogeneity; genotypes F and H had leucine at this position while all other genotypes had alanine (*Figure 1—figure supplement 1*). Analysis using the HBVdb database revealed that this leucine mutation is present at a very low frequency (about 3.75%) among all reported HBV sequences (total 7407 sequences) (*Hayer et al., 2013*), and is almost completely limited to genotypes F and H (*Figure 2—figure supplement 5B*). ELISA binding assays showed that 2H5-A14 bound weakly to a preS1 peptide variant containing the epitope-forming seqnces of both the F and H genotypes of HBV (*Figure 2—figure supplement 5C*), suggesting that 2H5-A14 likely has limited efficacy against these rare genotypes. Even so, our sequence analysis showed that the epitope 2H5-A14 is highly conserved and our neutralization assay results clearly showed that 2H5-A14 neutralizes the three major epidemic genotypes of HBV with remarkable potency. Given this, it is reasonable to extrapolate that 2H5-A14 can likely neutralize the vast majority (>95%) of HBV virus strains.

## 2H5-A14 prevented HDV infection in a mouse model with chimeric NTCP

We evaluated the prophylactic efficacy of 2H5-A14 against HDV infection in a recently established gene-edited mouse model in which residues 84–87 of mNTCP have been replaced by their hNTCP counterparts (*He et al., 2016*). 2H5-A14, HBIG or an isotype-matched control antibody was administered at indicated doses to *Ntcp* gene-edited homozygous FVB mice 1 hr prior to HDV challenge. A 15 mg/kg dose of 2H5-A14 completely blocked HDV infection, reducing HDV titers in liver to below their limit of detection at day six after HDV challenge. Notably, 3 mg/kg and 0.6 mg/kg doses reduced HDV titers by 99.4% and 98.5%, respectively. In contrast, even a high dose of HBIG (100 IU/kg) only reduced HDV titers by 90%, which was still insufficient to prevent HDV infection (*Figure 3A*).

## Prophylactic and therapeutic efficacy of 2H5-A14 in a liver-humanized mouse model of HBV infection

We next examined prophylactic efficacy of 2H5-A14 against HBV infection in human liver chimeric mice. Chimeric livers generated by transplanting human hepatocytes into *Fah*[-/-]*Rag2*[-/-]*/IL2rg*[-/-] (FRG) triple knock-out mice ('hFRG mice') are known to be highly susceptible to HBV infection (*Strom et al., 2010*; *Bissig et al., 2010a*; *Azuma et al., 2007*). We tested the in vivo efficacy of 2H5-A14 in neutralizing viral challenge both before (prophylactically) and after (therapeutically) administration of patient-derived viruses to these chimeric mice. The hFRG mice used in this

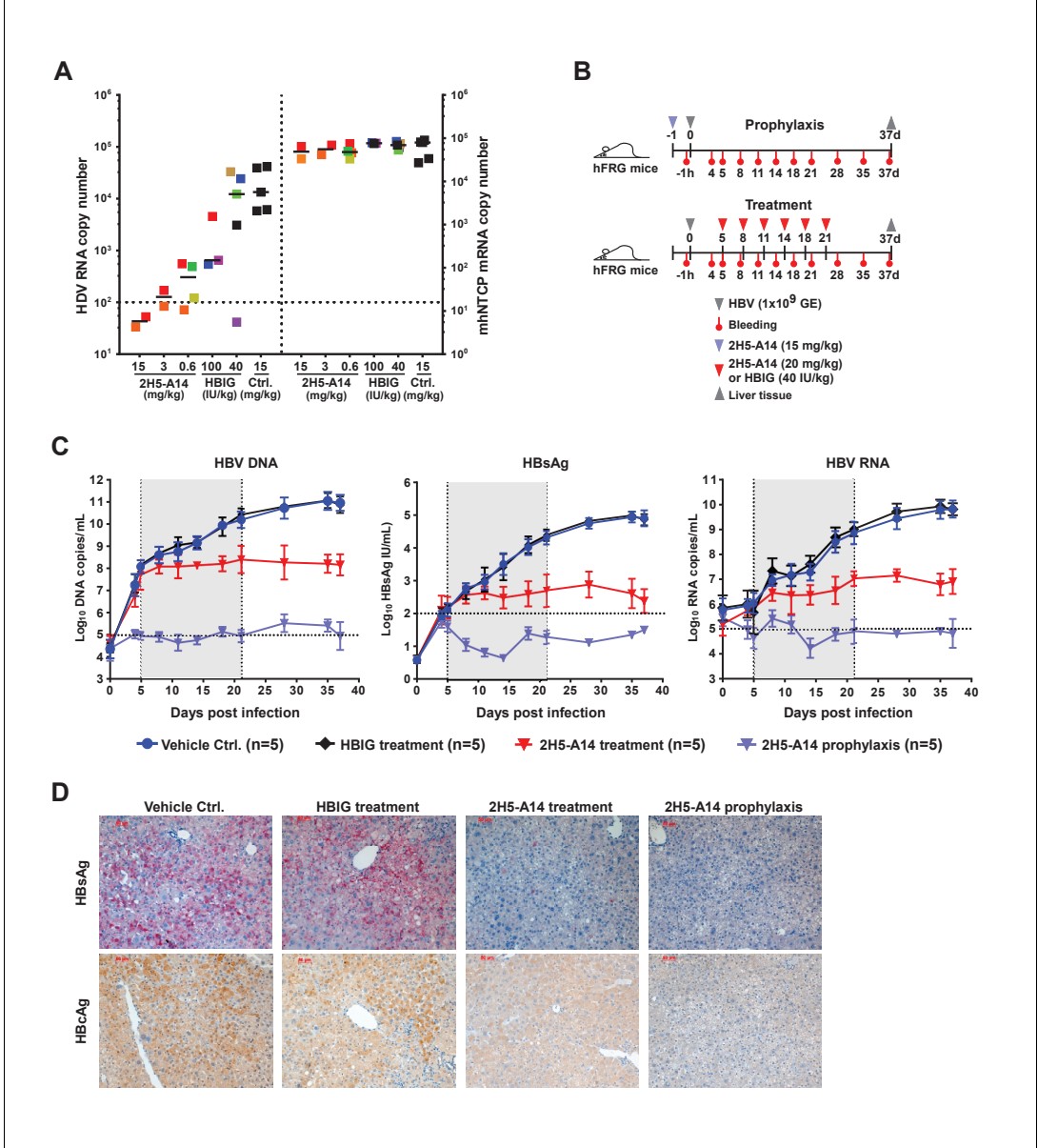

**Figure 3.** Prophylactic and therapeutic efficacy of 2H5-A14 in mouse models. (**A**) 2H5-A14 protected NTCP-edited mice from HDV infection. A total of 21 mice (NTCP-edited homozygous FVB) were IP administered 2H5-A14, HBIG, or matched isotype control Ab at the indicated concentrations at 9 days after birth. One hour later, each mouse was challenged with $1 \times 10^{10}$ GE of HDV and was then sacrificed on post infection day 6. The liver tissues were collected from each mouse. HDV total RNA levels and the edited NTCP gene expression were measured by qPCR. The data were presented similarly as in *Figure 1F*. (**B**) Schematic diagram illustrating HBV challenging, bleeding and antibody administration schedules in the mouse study. Five animals were used in each group. Recipient hFRG mice were challenged with $1 \times 10^{9}$ GE HBV (Genotype B) on day 0. A single dose of 15 mg/kg 2H5-A14 was IP administered one day before viral challenge (prophylaxis), or biweekly doses of 20 mg/kg 2H5-A14 were IP administered starting from day 5 to day 21 after the challenge (therapeutic). HBIG (40 IU/kg) treatment was used as a control. (**C**) Levels of HBV DNA and RNA, and HBsAg concentrations in mouse serum. Blood samples from all the mice (panel B) were collected at the indicated time points for measuring HBV DNA, HBV RNA, and HBsAg levels. The horizontal dotted lines indicate the reliable detection limits; the vertical dotted lines and the grey-shaded areas indicate the treatment window. (**D**) IHC staining of HBsAg and HBcAg in liver tissues from sacrificed mice at the end of the experiment. Intrahepatic HBsAg and HBcAg levels were detected by specific antibodies conjugated with Alkaline phosphatase (AP) and horseradish peroxidase (HRP), respectively. Each set of images represents the results for one mouse of each group.

DOI: https://doi.org/10.7554/eLife.26738.015

The following source data and figure supplements are available for figure 3:

**Source data 1.** Data for *Figure 3A and 3C*.

DOI: https://doi.org/10.7554/eLife.26738.019

*Figure 3 continued on next page*

*Figure 3 continued*

**Figure supplement 1.** Human albumin level and ALT activity in hFRG mouse serum samples.
DOI: https://doi.org/10.7554/eLife.26738.016
**Figure supplement 2.** Intrahepatic HBV total DNA, RNA, and cccDNA in liver tissues of HBV-infected hFRG mice.
DOI: https://doi.org/10.7554/eLife.26738.017
**Figure supplement 3.** Intrahepatic HBsAg and HBcAg in liver tissues of HBV-infected hFRG mice.
DOI: https://doi.org/10.7554/eLife.26738.018

experiment had high level of humanization with an average human serum albumin level of about 10 mg/mL. This experiment also included virus infection only (Vehicle Ctrl.) and HBIG control group. All of the treatments began at five dpi and lasted until 21 dpi (*Figure 3B*). Secreted HBsAg, as well as HBV DNA, and RNA levels ('virological markers') were measured periodically from serum samples. HBV rapidly and robustly replicated in the hFRG mice (*Figure 3C*). In the infection-only control group, the serum HBV DNA and RNA levels reached about $1 \times 10^{11}$ and $1 \times 10^{10}$ copies/mL, respectively, and HBsAg level exceeded $1 \times 10^5$ IU/mL. A single administration of 2H5-A14 (15 mg/kg) one day prior to HBV challenge completely protected mice from HBV infection, as indicated by the levels of all of the virological markers in serum samples. Therapeutically, biweekly 2H5-A14 treatment (20 mg/kg) greatly reduced all of the virological markers in sera as compared to infection-only control and to the HBIG treatment groups. Remarkably, by the end of the experiment (37 dpi), 2H5-A14 treatment resulted in three log reductions in the HBV DNA and RNA levels relative to the two control groups (*Figure 3C*). In addition, human albumin levels and alanine aminotransferase (ALT) activities in mouse serum samples were monitored throughout the study period. There were no significant differences between 2H5-A14 treated group and other groups including vehicle control, 2H5-A14 prophylaxis and HBIG control groups (*Figure 3—figure supplement 1*).

Liver tissues collected at the end of the experiment were used for immunohistochemical staining (IHC) of HBsAg and HBcAg, Southern blotting of viral DNA and cccDNA, and Northern blotting of viral RNA. Consistent with our sera-based measurements, liver tissues from 2H5-A14 prophylaxis group animals had no detectable HBsAg and HBcAg or HBV DNA, cccDNA, or RNA; while both the infection vehicle control and HBIG treatment groups had strong signals for these markers; liver tissues from 2H5-A14 treatment had greatly reduced HBsAg and HBcAg, and had almost undetectable levels of HBV DNA and RNA (*Figure 3D* and *Figure 3—figure supplements 2–3*).

## 2H5-A14 mediated ADCC and ADCP but not CDC

Encouraged by the therapeutic effects observed for 2H5-A14 in the hFRG mouse model, we further dissected the mechanism-of-action of 2H5-A14. To determine whether viral neutralization mediated by Fab of 2H5-A14 is the sole anti-viral mechanism or if, additionally, effector functions mediated by the Fc region may be involved, we first examined the effector functions of 2H5-A14 using in vitro assays.

Human NK cells and macrophages are two types of immune effector cells capable of mediating ADCC and ADCP. NK cells primarily express FcγRIIIa (CD16) and mediate ADCC, whereas macrophages express all classes of FcγRs and can perform both ADCC and ADCP (*Braster et al., 2014*; *Herter et al., 2014*; *Bournazos et al., 2015*). We initially tested whether 2H5-A14 can activate NK cells to induce ADCC. These experiments used an in vitro ADCC assay that employed human natural killer cells expressing both the V158 allele of FcγRIIIa and the γ chain of the Fc receptor (NK92-MI^hCD16^ line) as effectors (*Klingemann et al., 2016*). Two stable CHO cell lines expressing either the wild type (WT) epitope or a mutant epitope of 2H5-A14 were established: CHO-59C cells expressing a recombinant single-transmembrane protein (WT preS1 fused with the transmembrane domain of VAMP2, vesicle-associated membrane proteins), to which 2H5-A14 efficiently binds, was used as the target cell line; CHO-59C-mut cells carrying a D20A mutation in preS1 that abolishes 2H5-A14 binding was used as the control cell line (*Figure 4—figure supplement 1A*). 2H5-A14 but not an isotype-matched control antibody treatment resulted in a dose-dependent cytotoxic effect against CHO-59C cells; no such effect was observed for the CHO-59C-mut control cells, demonstrating that NK cell-mediated ADCC requires the presence of the 2H5-A14 epitope on target cells (*Figure 4A*).

To further confirm the ADCC activity of 2H5-A14, a process that invariably requires Fc-FcγR interaction, we generated a 2H5-A14 variant (2H5-A14-DANA) with two mutations in its IgG1 Fc region

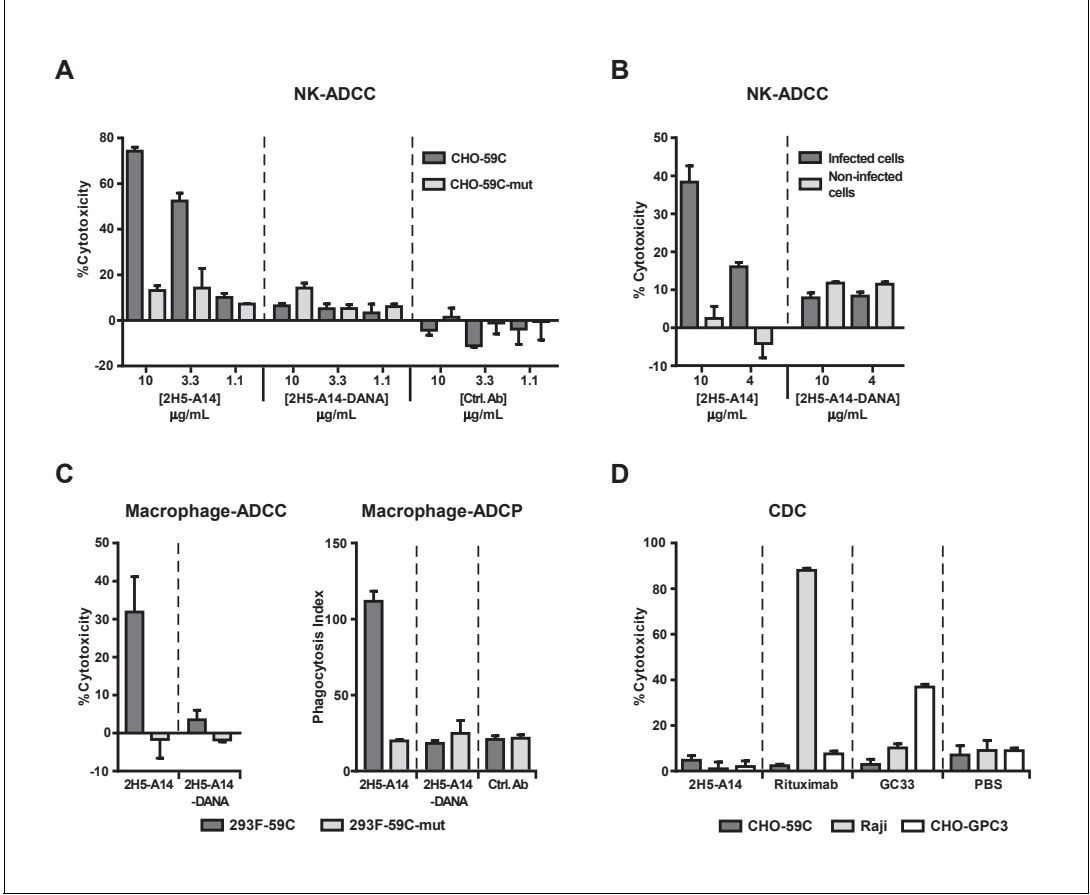

**Figure 4.** 2H5-A14 mediated ADCC and ADCP but not CDC. (**A–B**) 2H5-A14 induced ADCC via NK cells. NK92-MI[hCD16] was used as effector cells. 2H5-A14, its ADCC-inactive Fc variant (2H5-A14-DANA) or isotype-matched control Abs were tested at the indicated concentrations with three replicates. Different target cells were analyzed, including CHO cells expressing the epitope of 2H5-A14 inserted into the extracellular domain of VAMP2 on the cell surface (CHO-59C) or epitope mutant control cells (CHO-59C-mut) (**A**); the HBV-infected HepG2-hNTCP cells or non-infected cells (**B**). The ratio of effector cells to target cells (E:T) was 6:1. The ADCC activity was measured using a lactate dehydrogenase (LDH) release assay, showing as percentages of cytotoxicity. (**C**) 2H5-A14 induced ADCC and ADCP via macrophages. The target cells used were suspension 293F cells expressing the 2H5-A14 epitope or the mutant epitope, as described above for the CHO cells (**A**). The effector cells were human blood monocyte-derived macrophages. The E:T ratio was about 1:1. The ADCC activity (left) was measured as described above. For the ADCP assay (right), the target cells were labeled with CFSE fluorescent dye prior to mixing with macrophages. ADCP activity was monitored by fluorescence microscopy. The phagocytosis index was determined as the number of CFSE-positive target cells per 100 macrophages. The means and error bars shown were from two independent experiments. (**D**) CDC activity of 2H5-A14 and other control antibodies. Rabbit complement sera (10%) (Sigma-Aldrich) were used. All the samples were tested in triplicates. Rituximab (anti-CD20) and GC33 (anti-GPC3) mAbs were positive control mAbs for the assay. They have the same Fc region of human IgG1 as 2H5-A14. All data shown in panel A, B and D represent at least two independent experiments. All the antibodies were tested at 10 μg/mL in panel C-D.
DOI: https://doi.org/10.7554/eLife.26738.020

The following source data and figure supplements are available for figure 4:

**Source data 1.** Data for *Figure 4A–D*.
DOI: https://doi.org/10.7554/eLife.26738.025
**Figure supplement 1.** Characterization of 2H5-A14 and its Fc mutant (2H5-A14-DANA) by FACS.
DOI: https://doi.org/10.7554/eLife.26738.021
**Figure supplement 2.** Characterization of the binding of 2H5-A14 and its Fc mutant (2H5-A14-DANA) with human or mouse FcγRs via Biacore analysis.
DOI: https://doi.org/10.7554/eLife.26738.022
**Figure supplement 3.** 2H5-A14-mediated ADCP via human and mouse macrophages.
DOI: https://doi.org/10.7554/eLife.26738.023
**Figure supplement 4.** CDC and complement C1q binding activity of 2H5-A14 and other antibodies.
DOI: https://doi.org/10.7554/eLife.26738.024

(D265A and N297A) that are known to abolish the binding of Fc with all classes of FcγRs (*Shields et al., 2001*; *Wilson et al., 2011*). As expected, the 2H5-A14-DANA variant could still bind to CHO-59C cells and preS1 peptides, and neutralize HBV infection to the same extent as 2H5-A14 (*Figure 4—figure supplement 1A–C*), but could not bind to Fc receptors as revealed by FACS binding assays using NK92-MI[hCD16] cells and SPR analysis (Biacore) using soluble human and mouse FcγRs (*Figure 4—figure supplement 1D* and *Figure 4—figure supplement 2*). Consistently, the 2H5-A14-DANA variant could not induce ADCC activity (*Figure 4A*). Using the same ADCC assay, we also demonstrated that 2H5-A14, but not the 2H5-A14-DANA variant, caused ADCC against HBV-infected cells. The HepG2-hNTCP cells infected by HBV were specifically killed by NK cells, whereas the HepG2-hNTCP cells uninfected by HBV were spared from NK cell killing (*Figure 4B*).

We next examined if 2H5-A14 can mediate ADCC and ADCP via macrophages. When fully differentiated human macrophages were mixed with target cells (293F-59C, see materials and methods) expressing the 2H5-A14 epitope on their surface, both ADCP and ADCC were observed following treatment with wild type 2H5-A14 but not with the 2H5-A14-DANA variant (*Figure 4C*, *Figure 4—figure supplement 3A*). FACS analysis confirmed that 2H5-A14 and 2H5-A14-DANA bound to these target cells to a similar extent (*Figure 4—figure supplement 3B*). When the target cells were expressing a mutant epitope (293F-59C-mut), no specific ADCC and ADCP were observed (*Figure 4C*). Similar results were obtained using mouse bone marrow-derived macrophages (BMDMs) as effector cells (*Figure 4—figure supplement 3C*). Taken together, the results of these in vitro assays suggest that 2H5-A14 can kill target cells via Fc-FγR interaction-dependent effector functions such as ADCC and ADCP.

We further analyzed if 2H5-A14 can activate complement and induce CDC. In the presence of complement sera (from rabbit or human), 2H5-A14 did not cause cell lysis of CHO-59C cells. Under the same testing conditions, anti-CD20 (Rituximab) and an anti-GPC3 chimeric antibody (GC33) that are known to cause CDC all induced target-specific CDC (*Figure 4D* and *Figure 4—figure supplement 4A*) as previously reported (*Nakano et al., 2009*; *van Meerten et al., 2006*). Thus, 2H5-A14 is not competent to induce CDC against target cells. Consistently, 2H5-A14 has very weak binding activity to both human- and mouse C1q (*Figure 4—figure supplement 4B*).

## The therapeutic effect of 2H5-A14 against HBV infection is dependent on Fc-mediated effector functions

To assess whether the effector functions of 2H5-A14 contribute to its in vivo efficacy, we compared the anti-viral activities of 2H5-A14 and 2H5-A14-DANA in the aforementioned HDV and HBV infection mouse models. The Fc of 2H5-A14 is from the human IgG1 isotype, which is able to engage mouse FcγRs on mouse effector cells (*Figure 4—figure supplement 2*) (*Overdijk et al., 2012*). In the *Ntcp* gene-edited homozygous FVB mice, 2H5-A14-DANA and 2H5-A14 showed similar prophylactic efficacy: both offered complete protection from HDV infection at a 5 mg/kg dose (*Figure 5A*). This result suggests that Fab-mediated neutralization, rather than the Fc-mediated effector function, is primarily responsible for preventing viral infection. This mouse model only supports transiently-acute HDV infection and is thus only suitable to assess prophylactic efficacy. We therefore used the hFRG mouse model, which supports sustained HBV infection, to determine if the therapeutic effect of 2H5-A14 requires Fc-FγR interaction-dependent effector functions. hFRG mice with a medium level of humanization (an average human serum albumin level of about 3 mg/mL) were challenged with HBV on day 0. At 33 dpi, nine mice with similar level of stably-established HBV infection were divided into three groups and treated with 2H5-A14 (5 mg/kg), 2H5-A14-DANA (5 mg/kg), or PBS (vehicle control). Treatment was administered twice weekly for a total of 4 weeks (*Figure 5B*). Blood was collected periodically for monitoring virological markers, including DNA titer and the HBsAg level until the end of the experiment (89 dpi). The HBV DNA titer and the HBsAg level increased over time in the control group, with an average DNA titer of ~$2.0\times10^9$/mL and an average HBsAg level of ~$3\times10^6$ IU/mL at 89 dpi. In the 2H5-A14-DANA group, the average HBV DNA titer and the HBsAg levels were ~$3.8\times10^8$/mL and ~$4.0\times10^4$ IU/mL, respectively, at 89 dpi. In contrast, compared to the control group, the 2H5-A14 treatment group had an about 200-fold reduction in DNA titer (~$1.0\times10^7$/mL) and a ~ 40000 fold reduction in the HBsAg level (~70 IU/mL) at 89 dpi. Importantly, in contrast to control PBS or 2H5-A14-DANA, treatment with 2H5-A14 resulted in reductions in both HBV DNA titer and HBsAg levels over time, starting from the beginning of the treatment and continuing through the treatment window and 3 weeks after treatment withdrawal, with only a slight

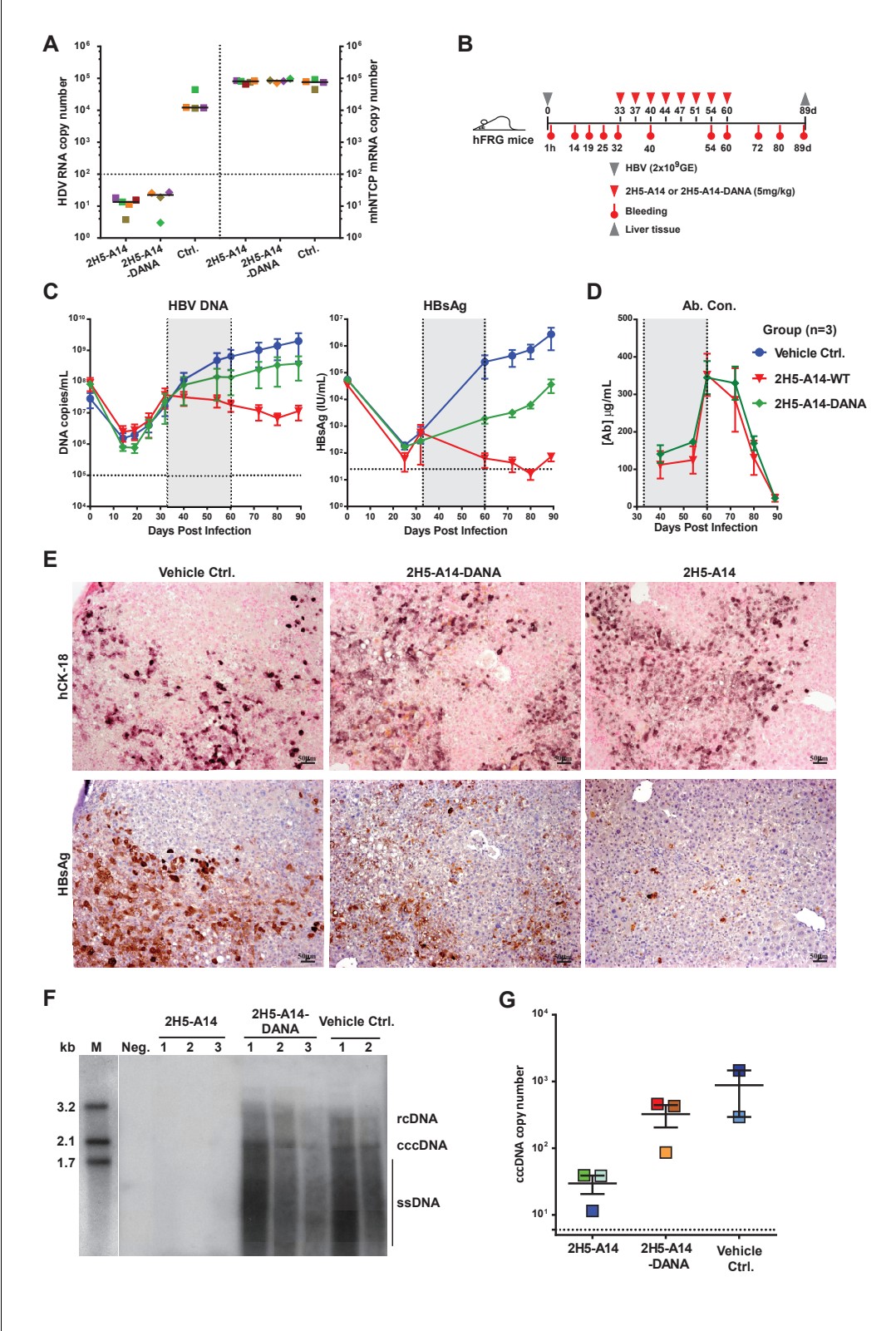

**Figure 5.** Contributions of Fc-mediated effector functions to the therapeutic effect of 2H5-A14 against HDV and HBV infection in mouse models. (**A**) Fc-mediated effector functions are not required for 2H5-A14 to protect mice from HDV infection. The experiment was performed similarly to that described in *Figure 3A*. Both 2H5-A14 and Fc mutant 2H5-A14-DANA were tested at 5 mg/kg. The copy numbers shown in the Y-axes are from 20 ng of total liver RNA for each sample. (**B**) Schematic diagram illustrating HBV challenge, bleeding and antibody treatment schedules in the mouse study.
*Figure 5 continued on next page*

*Figure 5 continued*

Three animals were used in each group. Recipient hFRG mice were challenged with HBV viruses (genotype D) at $2 \times 10^9$ GE. Twice weekly treatment with PBS, 2H5-A14 (5 mg/kg), or 2H5-A14-DANA (5 mg/kg) began at 33 dpi and lasted for 4 weeks. (C) HBV DNA titers and HBsAg levels in serum. Blood samples were collected at the indicated time points for measuring HBV DNA titers and/or HBsAg levels. (D) Antibody concentrations. The antibody concentrations in serum samples were measured by ELISA using antibody standards with known concentrations. Note, for dpi 40 and dpi 54, the blood samples were collected at three days after Ab administration; for dpi 60, the blood samples were collected 1 hr after Ab administration. The horizontal dotted lines indicate the lowest detection limits; the vertical dotted lines and the grey-shaded areas indicate the treatment window (C–D). (E) IHC staining of human cytokeratin-18 (hCK18) and HBsAg in serial sections of liver tissues from sacrificed hFRG mice at the end of the experiment (dpi 89). Intrahepatic HBsAg was detected by a specific anti-HBsAg mouse mAb, followed by HRP-anti-mouse secondary Ab and stained brown using DAB substrate, nuclei were stained blue by Hematoxylin. Human hepatocytes in consecutive tissue sections were visualized by staining with a human-specific hCK18 mAb and DAB substrate in blue violet, nuclei were stained with Nuclear Fast Red. Each image shown represents the staining result for one mouse of each group. (F) Southern blot analysis of intrahepatic viral DNA. Total HBV DNA was extracted from mouse liver tissues at dpi 89. The extracted DNA samples were analyzed by Southern blotting with an $^{[\alpha\text{-}32P]}$dCTP-labeled full-length HBV DNA probe. DNA samples prepared from normal mouse livers were used as a negative control. 100 pg each of 3.2 kb, 2.1 kb, and 1.7 kb HBV DNA fragments were used as DNA markers. rcDNA (relaxed circular DNA), cccDNA, and ssDNA (single-strand DNA) intermediates are labeled. (G) qPCR quantification of intrahepatic viral cccDNA level. 500 ng of total DNA prepared from the liver tissues collected at dpi 89 was digested by PSAD and 1/10 of the digested samples were used to quantify HBV cccDNA by qPCR using specific primers (see Materials and methods). The hNTCP gene copy numbers, which represent the amount of human hepatocytes in the liver tissues of chimeric mice, were used to normalize the cccDNA level in the human hepatocytes for each sample. The cccDNA copy number shown in the Y-axis is the relative value normalized to 1000 copies of hNTCP gene in ~50 ng total DNA samples. Each square represents one mouse. The horizontal dotted line indicates the reliable detection limit.

DOI: https://doi.org/10.7554/eLife.26738.026

The following source data and figure supplements are available for figure 5:

**Source data 1.** Data for *Figure 5A, 5C-D, and G*.
DOI: https://doi.org/10.7554/eLife.26738.029
**Figure supplement 1.** Human albumin levels in hFRG mouse serum and IHC of HBsAg in liver tissues of HBV-infected hFRG mice.
DOI: https://doi.org/10.7554/eLife.26738.027
**Figure supplement 2.** Phylogenetic trees of the protein sequences of the L gene cloned from the liver tissues of HBV-infected hFRG mice.
DOI: https://doi.org/10.7554/eLife.26738.028

rebound at the end of the experiment (4 weeks after treatment withdrawal) (*Figure 5C*). These results showed that 2H5-A14 treatment not only efficiently suppressed the spread of HBV in mice, but could also gradually reduce the extent of an established HBV infection over time. In contrast, 2H5-A14-DANA was able to suppress the spread of HBV as compared to the control group, but it did not reduce the extent of an established HBV infection: the levels of HBV DNA and HBsAg at all tested time points after the treatment were all higher than those at the starting point of treatment. Note that both 2H5-A14 and 2H5-A14-DANA treated mice had comparable serum antibody concentrations at all time points tested (*Figure 5D*) and had comparable human albumin levels both before viral challenge and at 80 dpi (*Figure 5—figure supplement 1A*).

Liver tissues collected at the end of the experiment were used for IHC staining of HBsAg and Southern blotting of viral DNA and cccDNA. Consistent with our sera-based measurements, liver tissues from 2H5-A14-treated but not 2H5-A14-DANA-treated mice had greatly reduced levels of HBsAg (IHC assay) comparing to the control mice (*Figure 5E* and *Figure 5—figure supplement 1B*). No significant histopathological changes related to the treatment were observed. Southern blotting showed that the HBV total DNA and cccDNA were undetectable for the 2H5-A14-treatment group, whereas strong total DNA and cccDNA signals were observed for the 2H5-A14-DANA and control groups (*Figure 5F*). Quantification of the cccDNA level in the liver tissues by qPCR further independently confirmed that 2H5-A14-treated mice indeed had much lower cccDNA copy numbers than the other two groups (*Figure 5G*). These results demonstrate the potent in vivo anti-viral effect of 2H5-A14. Moreover, the comparison of the effects of 2H5-A14 and 2H5-A14-DANA treatments side by side clearly proves that the anti-viral effect of 2H5-A14 results only in part from Fab-mediated neutralization, while the Fc-dependent effector functions of 2H5-A14 are particularly critical in conferring its overall therapeutic effect.

## Prospects for immune escape

To examine if HBV escape mutants will arise under selection pressure from 2H5-A14 treatment, we sequenced the HBV *L* gene amplified from the mouse liver tissues collected at the end of the two

aforementioned studies in hFRG mice. Few random mutations were observed within the L genes in both mouse studies, and it appeared that more mutations were in the 2H5-A14 treatment group than that in the control group, but no mutation was specifically associated with 2H5-A14 treatment and no mutations were found within the antibody's epitope (*Figure 5—figure supplement 2*). Although we cannot completely rule out the possibility that 2H5-A14 treatment may induce escape mutants, these results indicate that the epitope of 2H5-A14, a receptor-binding proximal and highly conservative region centered by Phe[23], is likely unsusceptible to escape mutation.

## Discussion

Highly potent direct anti-viral agents for chronic hepatitis C (HCV) can now cure most HCV-infected patients (*Schreiber et al., 2016*; *Feld et al., 2015*). Curing HBV has thus become the next major goal in fighting viral hepatitis. Monoclonal antibodies have rapidly become an important drug class, especially for treating cancer and autoimmune diseases. However the potential of using mAbs to treat chronic viral infections is only beginning to be explored.

In HBV, nAbs against the HBV small (S) envelop protein (HBsAg) constitute the major protective components of both the widely-used recombinant HBV vaccine and of the post-exposure prophylactic blood-derived HBIG. However, neither the vaccine nor HBIG are effective in treating persistent HBV infection (*Tsuge et al., 2016*). A previous study reported that a combination of two monoclonal nAbs against HBsAg (HepeX-B) only exhibited short-term anti-viral effects in animal models and an early phase I human study (*Eren et al., 2000*; *Neumann et al., 2010*). A humanized anti-HBsAg nAb was recently shown to have longer anti-viral effects in mouse models mainly through clearance of the Ab-HBsAg immune complexes by phagocytosis (*Zhang et al., 2016*). High levels of circulating HBsAg-containing non-infectious subviral particles are commonly present in the blood of chronically-infected HBV patients (*Seeger et al., 2007*). It has been assumed that HBsAg can act as a decoy that prevents anti-HBsAg nAbs from effective targeting of infectious virions and infected cells, the formation of circulating immune complexes that may lead to immune complex disease also remains a potential risk. Compared to the HBsAg, the L protein is much less abundant in circulation and preferentially present on infectious HBV particles (*Seeger et al., 2007*). Moreover it is now known that the preS1 domain of the L protein is responsible for receptor NTCP binding (*Yan et al., 2012*).

To date, two anti-preS1 nAbs, the murine BX-182 nAb and humanized KR127 nAb, have been shown to be able to prevent HBV infection when injected in chimpanzees prior to HBV challenge (*Chi et al., 2007*; *Zhang et al., 2006*). However, neither these nor any other anti-preS1 nAbs have been tested as a treatment against established HBV infection and no mechanistic studies have been conducted with these nAbs. BX-182 is an HBV subtype-specific nAb owing to sequence variations in its epitope (residues 6–10 of preS1, SVPNP). KR127 recognizes residues 37–45 (NSNNPDWDF) of preS1 and has broader neutralizing activity than BX-182, but its potency is limited, diminishing its promise for therapy.

2H5-A14 has potent viral neutralization activity with low picomolar $IC_{50}$ values in cell cultures. Consistent with this in vitro potency, it is extremely effective in protecting mice from de novo HDV and HBV infection. Remarkably, 2H5-A14 demonstrated impressive therapeutic efficacy against HBV infection in two independent mouse studies using liver-humanized hFRG mice. At present, there is no immune-competent small animal model that supports natural and long-lasting HBV infection. Although the hFRG mice are immunodeficient (*Azuma et al., 2007*), this model recapitulates the complete life cycle of HBV and is a valuable model for evaluating the efficacy of an antibody against HBV infection in vivo. Our initial animal study in hFRG mice with high-level of humanized chimeric liver was ended at 37 days due to the rampant viral propagation in the untreated control group, which had an adverse effect on the general health of these mice. The second study had slower viral propagation rates (likely due to the medium-level of liver humanization and/or the relatively lower propagation rate of the challenging virus) and lasted for 89 days, allowing us to assess the antiviral effect of 2H5-A14 over a longer period. Monitoring infection during this longer experimental period revealed a marked reduction of the extent of an established HBV infection in 2H5-A14-treated mice.

Mechanistically, we found that viral neutralization by 2H5-A14 is the dominant mechanism contributing to the protection of mice from HDV infection. In contrast, in addition to protecting hepatocytes from viral entry, Fc-mediated effector functions contribute substantially to the overall antiviral effect of 2H5-A14 against HBV infection in hFRG mice. Our experiments in the hFRG model revealed

that wildtype 2H5-A14, but not the Fc-DANA mutant of 2H5-A14 (a variant incapable of mediating antibody-dependent effector functions), elicited remarkable anti-viral effects against an established HBV infection. hFRG mice lack T and B lymphocytes, and NK cells. NK cells are generally considered to be the main effector cells in mediating ADCC to induce apoptosis in target cells (*Wang et al., 2015*). The observed effector functions in hFRG model thus cannot be attributed to NK cells but to the remaining immune effector cells in these mice; for example FcγR-expressing macrophages or other meloid cells (e.g. neutraphils) that are capable of killing infected hepatocytes by ADCC and/or ADCP or antibody-dependent scretion of antiviral cytokines or by direct clearance of the antibody-opsonized viruses via phagocytosis. Considering that primary hepatocytes and HBV-infected hepatocytes may be resistant to CDC (*Halme et al., 2009*; *Koch et al., 2005*; *Liu et al., 2013*; *Zhang et al., 2013*), and 2H5-A14 lacked complement C1q binding activity and CDC activity in vitro, CDC is unlikely involved in the anti-viral effect of 2H5-A14 in the HBV-infected hFRG mice. Macrophages have previosuly been shown to be the major effector cells invovled in anti-tumor activity of several mAbs in mice (*Clynes et al., 2000*; *Hamaguchi et al., 2006*; *Minard-Colin et al., 2008*). Macrophages exert their cytotoxic functions via diverse mechanisms such as ADCC, ADCP, etc. (*Braster et al., 2014*). Neutrophils have also been reported to be involved in the anti-tumor efficacy of two mAbs in tumor models in mice (*Siders et al., 2010*; *Golay et al., 2013*). It will be interesting to identify which immune effector cells are involved in mediating the in vivo antiviral effect of 2H5-A14 in future studies. The cells involved could be identified via the selective depletion of macropahges (*van Rooijen and Hendrikx, 2010*) or neutraphils (*Daley et al., 2008*) using specific reagents targeting each of them in hFRG mice.

In humans, the mechanisms underlying HBV control is only partially understood, and harnessing immune control to combat chronic HBV infection is promising but remains a challenge (*Liang et al., 2015*; *Block et al., 2013*; *Zhang et al., 2015*; *Guidotti et al., 2015*; *Isorce et al., 2015*; *Lucifora et al., 2014*; *Bertoletti and Rivino, 2014*; *Bertoletti and Ferrari, 2012*). During chronic HBV infection, it appears that NK cells have impaired secretion of antiviral cytokines (e.g. IFN-γ) yet are able to maintain their cytotoxicity (*Maini and Peppa, 2013*; *Sun et al., 2015*; *Shabani et al., 2014*). The proportion of FcγR-expressing NK cells in liver (27–75% of total NK cells in liver) is lower than that in blood (*Burt et al., 2009*). Nontheless, given that they retain their cytotoxicity, these FcγR-expressing NK cells can be assumed to have ADCC activity. Hepatic macropahges include both infiltrating macrophages and liver-resident macrophages (Kupffer cells, accounting for 80–90% of all tissue macrophages in entire body [*Jenne and Kubes, 2013*]). The roles of these cells in HBV infection appear to be complicated (*Ju and Tacke, 2016*). Similar to human tumor-associated macrophages, hepatic macropahges likely retain antibody-dependent effector fuctions via their expressed FcγRs (*Herter et al., 2014*; *Grugan et al., 2012*). Future human clinical trials will allow us to determine if, as we found in hFRG mice, 2H5-A14 elicits its Fc-dependent effect functions via activating immune cells in liver to exert its anti-viral effect.

Two aspects of the anti-HBV effect of 2H5-A14 are worth noting. First, 2H5-A14 treatment resulted in substantial decreases in HBsAg levels in the blood of HBV-infected hFRG mice over time. Serological HBsAg reduction, and eventually HBsAg loss, is considered as a critical step toward a 'functional cure'. Current NUCs therapy rarely reduces HBsAg level in CHB patients. 2H5-A14 specifically targets preS1 but not the vast majority of HBsAg particles, therefore, in contrast to nAbs targeting HBsAg, the decline of HBsAg by 2H5-A14 cannot be attributed to the Ab-mediated clearance of HBsAg from the blood, but to a gradual reduction of established HBV infection. Second, 2H5-A14 treatment at least effectively suppressed cccDNA accumation and more likely reduced cccDNA in liver tissues. Due to technical difficulties, we cannot quantify the intrahepatic cccDNA when treatment began, however the continuing reduction of the levels of DNA titers and HBsAg levels after treatment clearly demonstrates that 2H5-A14 does reduce the extent of an established HBV infection. Our data showed that 2H5-A14 can mediate immune effector cells to kill HBV-infected cells in vitro via ADCC and ADCP, although more evdience is needed, it is conceivable to speculate that the clearance of cccDNA-carrying infected cells or decrease their in vivo half-lives by Fc-dependent effector functions via FcγR engagement, including cytolytic and/or noncytolytic mechanisms, may have contributed to the anti-HBV effect of 2H5-A14 in a manner beyond Fab-mediated blockage of new infections. It will be interesting to further determine the detailed mechanism of the antiviral effect of 2H5-A14, in particular the relative contribution of cytolytic versus noncytolytic effects in this process.

In summary, we identified six anti-preS1 human nAbs from a phage library and then improved the lead hit into 2H5-A14, a high-affinity and broad-spectrum nAb that is extremely potent in vitro and in vivo against both HDV and HBV. 2H5-A14 inhibits NTCP binding by recognizing a structurally-defined and highly-conserved epitope located in close proximity to the RBS of preS1. 2H5-A14 is also capable of triggering epitope-specific and Fc-dependent effector functions. Importantly, the effector functions contribute substantially to the anti-viral effect of 2H5-A14 against HBV infection in vivo. These results for the first time demonstrate a nAb targeting the RBS adjacent region on preS1 can exert anti-HBV effect through Fc-dependent effector functions by engaging FcγRs, pointing toward a novel antibody-Fc-dependent strategy for HBV treatment.

## Materials and methods

### Cell lines

Human embryonic kidney cell lines 293T (RRID:CVCL_0063) and human hepatocellular carcinoma cell line HepG2 (RRID:CVCL_0027) were from American Type Culture Collection (ATCC, Manassas, VA); human hepatocellular carcinoma cell line Huh-7 (RRID:CVCL_0336), Raji (RRID:CVCL_0511) and Chinese hamster ovary cell line CHO (RRID:CVCL_0213) were from the Cell Bank of Type Culture Collection, Chinese Academy of Sciences. HepG1-hNTCP cell line used for HBV and HDV in vitro neutralization assays were constructed as previously described (Sun et al., 2016). HepG2, Huh-7 and CHO cell lines were cultured with Dulbecco's Modification of Eagle's Medium (DMEM) supplemented with 10% fetal bovine serum, 100 U/ml penicillin and 100 μg/ml streptomycin. Raji cells were cultured with RPMI 1640 medium supplemented with 10% fetal bovine serum, 100 U/mL penicillin and 100 μg/mL streptomycin. These cells were cultured at 37°C in 5% CO2 humidified incubator. FreeStyle 293F cell line (RRID:CVCL_D603) was from Life Technologies, and the cells were cultured following manufacturer's instruction. The NK92-MI^hCD16 cell line was generously provided by Drs. Zhang and Li (Beigene). It was established by stably expressing FcγRIIIa (CD16, V158 allele) and FcRγ chain in the parental NK92-MI cell line (ATCC). All the cell lines used in this study had been tested for mycoplasma and that the tests were negative.

### Antibody library selection and screening of anti-preS1 antibodies

Two peptides derived from the pre-S1 domain of HBV genotype C were synthesized by Scilight-peptide (Beijing, China) with purity greater than 95%. NC36b, a peptide comprised of residues 4–36 of the pre-S1 domain of the HBV L protein with a biotin modification at its C-terminus. m47b, a lipopeptide comprised of amino acids 2–48 of the pre-S1 domain with a biotin modification at the C-terminus and a myristoylation modification at the N-terminus. A human non-immune scFv (single-chain fragment of variable domain: VH-linker-VL) antibody library ($1.1 \times 10^{10}$) constructed from peripheral blood mononuclear cells of 93 healthy donors was used for selection against the synthesized NC36b or m47b preS1 peptides. Briefly, the peptides were captured on streptavidin-conjugated magnetic M-280 Dynabeads (Thermo Fisher, Waltham, MA) and then incubated with $5 \times 10^{12}$ phage-scFv particles prepared from the library. For each peptide, two rounds of selection were performed. To recover high-affinity binders from the magnetic beads and to increase the diversity of the recovered phage-scFvs, two elution methods were used: peptide competition elution and conventional basic triethanolamine solution elution. Subsequently, a total of about 2000 single clones were randomly picked and rescued to produce phage-scFvs in the bacterial culture supernatant. These clones were screened for specific binding to NC36b and/or m47b by ELISA. More than 90% of the clones screened were positive for one or both peptides. Clones that bound to NC36b and/or m47b with an optical density at 450 nm >1.0 were selected for further antibody gene sequencing. The genes of the variable heavy chain (VH) and light chain (VL) of the selected clones were sequenced. A total of 109 clones with unique sequences were identified. These clones were either produced as purified phage-scFv particles or converted to full-length human IgG1s and then analyzed for binding to the preS1 peptides by ELISA again, and evaluated in HBV and/or HDV neutralization assays. At the end, top six clones were selected based on their binding and HBV/HDV neutralization activities.

## Preparation and purification of phage-scFvs for ELISA or neutralization assays

The phage-scFvs in the supernatant of 10–30 mL bacterial cultures were precipitated and concentrated with PEG/NaCl (*Sui et al., 2008*). The physical particle concentrations of the PEG-precipitated phage-scFvs were then quantified by spectrophotometry based on the UV absorption spectra of purified filamentous phages. The binding or viral neutralization activities of different phage-scFvs were then evaluated in ELISA assays or in cell-based viral neutralization assays, in a manner similar to the assays used for their full-length IgG1 form.

## Expression and purification of full-length human IgG1 antibodies

The coding sequences of the VH and VL of a scFv were subcloned, respectively, into a human IgG1 heavy chain (HC) expression vector and a light chain (LC) expression vector. To generate full-length hIgG1 antibodies, 293F or 293 T cells were co-transfected transiently with the two expression plasmids (HC +LC plasmids) at a 1:1 ratio. 3–5 days after transfection, the cell culture supernatant was harvested for purification of IgG1 via Protein A bead affinity chromatography.

## ELISA binding assay

1 µg/mL of streptavidin (Sigma Aldrich, St Louis, MO) in phosphate buffered saline (PBS) was coated onto U-bottom 96-well plates (Nunc, MaxiSorp, Denmark) at 100 µL per well, then incubated at 4°C overnight or 37°C for 1 hr. Biotin-labeled peptides (NC36b or m47b) at various concentrations in PBS (100 µL per well) were then captured onto the plates by incubating at 30°C for 1 hr. For phage-scFv based ELISA, serially diluted phage-scFvs in PBS containing 2% nonfat milk were added to each well at 100 µL per well. Specifically bound phage-scFvs were then detected by adding HRP-conjugated mouse anti-M13 antibody (GE Healthcare, Chicago, IL) and incubating for 1 hr at 30°C. After each incubation step, the ELISA plate was washed six times with PBS-T solution (0.05% Tween 20 containing PBS) at a volume of 300 µL of wash solution per well per wash. Following HRP-conjugated antibody incubation, the ELISA signal was developed with TMB substrate (Sigma) for 5–10 min at room temperature. The reaction was then stopped by adding 50 µL of 2M $H_2SO_4$ per well. The colorimetric absorbance was measured at 450 nm, with automatic subtraction of the reference absorbance at 630 nm by use of a Microplate Reader (Bio-Rad, Hercules, CA). For full-length hIgG1-based ELISA, the method was basically the same as described above for the phage-scFvs, with the exception that the bound antibodies were detected by HRP-conjugated mouse anti-human IgG Fc antibody (Pierce-Thermo Fisher Scientific, Waltham, MA).

## Competition ELISA assay

The competition ELISA assays were performed in a manner similar to the aforementioned binding ELISA assays, except that the tested Abs were incubated with captured peptide antigen in the presence of competition peptides. Briefly, different short peptides at serially diluted concentrations were mixed with 0.5 nM (75 ng/mL) of IgG1 antibody and added to the ELISA plates to compete for the binding of Abs to the captured peptides on the ELISA plates.

## Crystallization and structural determination of 2H5 and 59C complex

The amino acid sequence of 59C corresponds to aa (−10 ~48) of preS1 of genotype C. 2H5-scFv and 59C were co-expressed in *E. coli*. The complex was purified with Immobilized Metal Ion Affinity Chromatography using Ni-NTA agarose beads (QIAGEN, Germantown, MD), followed by size exclusion chromatography with a Superdex S200 10/300 column (GE Healthcare). The purified 2H5-scFv/59C complex was then concentrated and crystallized at 20°C using the hanging drop vapor diffusion method by mixing 1 µL of protein (29 mg/mL in 10 mM Tris-HCl pH 8.0 and 100 mM NaCl) and 1 µL of reservoir solution containing 2.8 M sodium acetate, pH 7.0. Needle-shaped crystals appeared after 10 days. The crystals were flash cooled in liquid nitrogen. The X-ray diffraction data were collected at beamline BL17U (*Wang et al., 2015*) of the Shanghai Synchrotron Radiation Facility and processed by HKL2000 (*Otwinowski and Minor, 1997*). The structure was determined at 2.5 Å resolution by molecular replacement in Phaser (*McCoy et al., 2007*; *McCoy, 2007*) using VH and VL derived from the structure of the Herceptin-Fab complex (PDB 3H0T) (*Jordan et al., 2009*) as a starting model. The initial model from molecular replacement was further refined in Phenix

(*Adams et al., 2010*) and manually rebuilt with Coot (*Emsley and Cowtan, 2004*). The final model includes 220 residues of 2H5 scFv and residues 20–27 of the 59C peptide. RAMPAGE analysis showed that 94.7% of the residues are in the favored region and 5.3% of residues are in the allowed region (*Lovell et al., 2003*).

## HBV and HDV neutralization assays

Recombinant HBV viruses of genotype D and HDV viruses with genotype D envelopes were produced by plasmid transfection of Huh-7 cells (*Yan et al., 2012*; *Sun et al., 2016*). The serum derived HBV viruses (genotype B and C) were from de-identified blood remnant samples of HBV patients. The neutralization assays were performed as previously described (*Yan et al., 2012*; *Yan et al., 2014*), with minor modifications.

For HBV infection of HepG2-hNTCP cells, the cells were cultured in PMM medium (*Yan et al., 2012*) for 24 hr in 48-well plates (~$5 \times 10^4$ cells/well) or in 96-well plates (~$2 \times 10^4$ cells/well) prior to viral infection. About 200 multiplicities of genome equivalents (mge) of HBV mixed with Abs were inoculated with HepG2-hNTCP cells in the presence of 5% PEG8000 and incubated for 16 hr. Cells were then washed three times with media and maintained in PMM. The cell culture medium was changed with fresh PMM medium every 2 days after the first medium change. At 3, 5, and seven dpi, the culture supernatants were collected and tested for HBV-secreted viral antigen HBsAg and/ or HBeAg with commercial ELISA kits (Wantai, China). The levels of HBeAg and/or HBsAg were used to evaluate the HBV neutralization activity of the antibodies. Data were normalized to the virus infection control and $IC_{50}$ values calculated from nonlinear regression curve fitting using GraphPad Prism six software.

For HBV neutralization assays using PHH cells, plasma-derived HBV (genotype B) were used. The neutralization assay was performed in 48-well plates with PHH cells harvested from humanized FRG mice. The PHH cells were infected with 200 mge of HBV in the presence of 2H5-A14, an isotype-matched control Ab, or HBIG at various concentrations. 16 hr after infection, the medium was changed to maintenance culture medium. Cell culture supernatants were harvested every 2 days for measuring HBsAg levels. The HBsAg level in serum samples was determined using a commercial kit (Autobio-CL0310, China). For HDV infection, the HepG2-hNTCP cells were infected similarly as with the HBV infection, but with about 500 mge of HDV viruses. Abs were mixed with viruses and then inoculated with the cells. At seven dpi, HDV-infected cells were fixed with 100% methanol at room temperature for 10 min. Intracellular delta antigen was stained with 5 μg/mL of FITC-conjugated 4G5 (a mouse anti-HDV Delta antigen monoclonal antibody) and nuclear delta antigen was stained with DAPI. Images were captured with a Nikon Eclipse Ti Fluorescence Microscope or a Zeiss LSM 510 Meta Confocal Microscope. The neutralization activity against HDV was determined based on the stained delta antigen amount (number of infected cells) and staining intensity.

## VH chain shuffling sub-library construction and selection

The VL gene of 2H5 was cloned into a phagemid vector containing a repertoire of non-immune VH genes (~$1 \times 10^{10}$) derived from 93 healthy donors. The constructed library had a size of $1.1 \times 10^9$. The library selection was done similarly as described above for antibody library selection of anti-preS1 antibodies. Only one round of selection was performed. A total of 576 individual clones were then randomly picked and screened by ELISA. Positive clones were sequenced to identify Abs with unique sequences.

## Binding kinetic analysis by Surface Plasmon Resonance (SPR) analysis

Kinetic analyses of 2H5-A14 or 2H5-A14-DANA mutant binding to FcγRs were performed on a Biacore T200 (Biacore, GE Healthcare) at 25°C similarly as described above. The FcγRs were expressed in 293F suspensions cells and purified by immobilized metal affinity chromatography using the $His_6$ tag fused to the C-terminus of FcγRs. ProteinA/G (Pierce, Thermo Fisher) was covalently attached to individual flow cell surfaces of a CM5 sensor chip by amine coupling using an amine coupling kit (Biacore). Antibodies at optimal concentrations were then captured on the chip to ensure the binding occurred as a homogenous 1:1 Langmuir interaction. The analyte FcγRs were then injected over each flow cell at serial diluted (2-fold) concentrations. A buffer injection served as a negative control. Upon completion of each association and dissociation cycle, surfaces were regenerated with 10 mM

pH2.0 Glycine-HCl solution. The association rates (ka), dissociation rates (kd), and affinity constants (KD) were calculated using BiacoreT200 evaluation software. For low affinity interactions, the starting FcγR concentration was from 4000 to 8000 nM, whereas for high-affinity interactions that was from 100 nM.

## Immunofluorescent analysis of 2H5-A14 competition of preS1 binding to hNTCP expressing cells

The FITC-labeled preS1 peptide, m59-FITC, was mixed with 2H5-A14 or control Ab at the indicated concentrations and then added to HepG2-hNTCP cells. The final concentration of m59-FTIC in the mixture was 100 nM. After incubation at 37°C for 15 min, the cells were washed five times with William's E medium. The DAPI-containing mounting solution was then added to the cells. The images were captured with a Zeiss LSM 510 Meta Confocal Microscope.

## ADCC assay using NK cells

The NK92-MI$^{hCD16}$ cells were used as effector cells in these assays; these cells express FcγRIIIa (CD16, V158 allele) and FcRγchain. Two types of cells were used as target cells: A CHO cell line (CHO-59C) stably expressing the epitope of 2H5-A14 (59C) and HBV-infected HepG2-hNTCP cells. Construction of CHO-59C cell line: An expression plasmid was first constructed by inserting the preS1-59C coding DNA into the C-terminal (CT) extracellular domain of the VAMP2 gene that encodes a single-pass type IV transmembrane protein. The NT of VAMP2 is on the cytoplasmic side of the membrane. Its CT extracellular domain only contains two amino acids. The 59C was added to the CT end of VAMP2; there was a 12aa long linker between 59C and VAMP2. The expression plasmid was then transfected into CHO cells, followed by FACS sorting of 2H5-A14-staining positive populations to obtain the stable cell line, CHO-59C. The CHO-59C-mut (D20A mutant in preS1-59C) stable cell line was similarly established and was used as a negative control cell line in the ADCC or ADCP assays. D20A mutant in preS1 knocked out the binding to 2H5-A14. m1Q mAb, which recognizes the D20A mutant form of 59C, was used for the FACS sorting of D20A mutant expressing cells. The HBV-infected cells were HepG2-hNTCP cells infected with HBV for 5 days before use in the ADCC assays.

For the ADCC assays, target cells were washed and resuspended in RPMI +5% FBS medium, and plated at 10000 cells/well in U-bottom 96-well plates and incubated briefly with various concentrations of 2H5-A14 or 2H5-A14-DANA. The effector cells were then added (60000 cells/well resulting in a raito of E:T = 6:1) to the wells containing the target cells and antibodies, and incubated for 6 hr at 37°C. Cytolysis was determined by lactate dehydrogenase (LDH) release following the instructions of a CytoTox 96 Non-Radioactive Cytotoxicity Assay kit (Promega, Fitchburg, WI). Percentage cytotoxicity was calculated following the manufacturer's instruction.

ADCC and ADCP by human macrophages. Human PBMCs were differentiated by 20 ng/mL macrophage colony-stimulating factor (M-CSF) (Peprotech, Rocky Hill, NJ) for 9 days before use as effector macrophages. Suspension 293F-59C and 293F-59C-mut cells were used as target cells for both ADCC and ADCP assays. These cells were similarly established as CHO-59C and CHO-59C-mut, except they were not stable transfectants but transiently-transfected cells. They were harvested and used as target cells at 30–48 hr after transient transfections. For both assays, the target cells (2 × 10$^5$ cells/well) were incubated with 10 μg/mL 2H5-A14-WT or 2H5-A14-DANA at RT for 10–30 mins and then added to the differentiated macrophages (~2×10$^5$ cells/well, resulting in a E:T ratio of 1:1) at 37°C for 2 hr. For ADCC, the cytolysis was determined by lactate dehydrogenase (LDH) release following the instructions of a CytoTox 96 Non-Radioactive Cytotoxicity Assay kit. For ADCP, the target cells were labeled with CFSE according to the manufacturer's protocol (Life Technologies) prior to incubation with antibodies. Phagocytosis of CFSE-labeled target cells by macrophages was recorded with a Zeiss Pascal Confocal System, and the phagocytic index was determined as the number of CFSE-positive cells per 100 macrophages. About 200 macrophages were counted per sample.

## ADCP by mouse macrophages

To prepare mouse bone marrow-derived macrophages (BMDMs), mouse bone marrow cells were flushed with a syringe from the tibia and femurs of C57BL/6 mice into DMEM medium. Cells were

collected and washed by PBS and filtered through a 40 µM cell strainer. To differentiate the cells into BMDMs, they were resuspended in DMEM medium supplemented with 15% L929 (secreting granulocyte/macrophage colony-stimulating factor, GM-CSF [*Englen et al., 1995*]) cell culture medium and cultured for 3 days without replenishing or changing medium. Suspension 293F-59C or 293F-59C-mut cells were used as target cells and labeled with CFSE. The differentiated BMDMs were labeled with a 1:200 dilution of anti-mouse F4/80-Alex Fluor647 (Thermo Fisher, clone BM8) prior to incubation with target cells. The CFSE labeled target cells were plated at a density of $2 \times 10^5$ cells/well and incubated with 10 µg/mL 2H5-A14-WT or 2H5-A14-DANA at RT for 15 mins and then added to the differentiated and labeled BMDMs ($\sim 2 \times 10^5$ cells/well, resulting in a E:T ratio of 1:1) at 37°C for 2 hr. Phagocytosis of CFSE-labeled target cells by macrophages was recorded with a Nikon A1R Confocal Microscope.

## Complement-mediated cytotoxicity (CDC)

Target cells were washed and resuspended in RPMI medium (without FBS) containing testing antibodies (10 µg/mL) and plated in a 96-well U-bottom plate at $2.5 \times 10^4$ cells/well. Complement sera derived from rabbit or human (Sigma-Aldrich) were then added to the wells containing the target cells and antibodies, the final sera concentration was 10%. After 2 hr (for rabbit serum) or 4 hr (for human serum) incubation at 37°C, the supernatants in each well were recovered and detected LDH release using a CytoTox 96® Non-Radioactive Cytotoxicity Assay kit (Promega). The cytolysis was calculated following the instructions of the kit. Rituximab or GC33 were used as systemic positive controls. Rituximab (anti-CD20) is commercial antibody used in clinic for treating lymphoma. GC33, a chimeric antibody against GPC3, was expressed and purified using the same method as described for full-length human IgG1 antibody. They both are capable of inducing CDC against their target cells (*Nakano et al., 2009*; *van Meerten et al., 2006*). 2H5-A14 has the same Fc region of human IgG1 as Rituximab and GC33.

## Prophylactic efficacy studies of 2H5, 2H5-A14 and 2H5-A14-DANA mutant against HDV infection in mouse models with humanized NTCP.

Human NTCP transgenic C57BL/6 mice (*hNTCP-Tg*) were used for evaluating 2H5 Ab (*Figure 1F*). The NTCP-edited FVB mice (residues 84–87 of mNTCP replaced by their human NTCP counterparts) were used for evaluating 2H5-A14 (*Figure 3A*) and for comparing the effect of 2H5-A14 and 2H5-A14-DANA (*Figure 5A*). For all experiments, homozygotes at 8–9 days after birth were used. Animals were hosted in the animal facility of NIBS, Beijing. The experiments were conducted following the National Guidelines for Housing and Care of Laboratory Animals and performed in accordance with institutional regulations after approval by the IACUC at NIBS.

Antibodies at the indicated concentrations were administrated 1–2 hr prior to HDV viral challenge. At day six after HDV challenge, mice were sacrificed and liver tissues were harvested and frozen liquid nitrogen immediately. The liver tissues were then homogenized and lysed. TRIZOL reagent was used to extract total RNA. The RNA samples were reverse transcribed into cDNA with a Prime Script RT-PCR Kit (Takara, Japan). cDNA obtained from 20 ng of RNA from each sample was used as the template for qPCR to quantify both the number of total HDV RNA and hNTCP mRNA copies with ABI Fast 7500 real time system instrument (Applied Biosystems, Waltham, MA) (*He et al., 2015*). The GAPDH mRNA was also quantified and used as an internal reference control for the qPCR assays. The HDV total RNA qPCR primers are: HDV-1184F: 5'-TCTTCCTCGGTCAACC TCTT-3'; HDV-1307R: 5'-ACAAGGAGAGGCAGGATCAC-3'. The hNTCP transgene qPCR primers are: forward primer: 5'-GCTTCTCCTCATTGCCATATTTT-3'; backward primer: 5'-GGGAGCAGTCC TCCCCT-3'. The edited chimeric m-hNTCP qPCR primers are: forward primer: 5'-GGTCTTTCGGC TGAAGAAC-3'; backward primer: 5'-CATGGCCAGGGTGAAGAGG-3'. The HDV RNA or NTCP mRNA copies were calculated with standard curves generated from samples with known copy numbers.

## Prevention and treatment of HBV infection in an hFRG mouse model

A mouse HBV infection model was previously established using FRG (*Fah⁻/⁻Rag2⁻/⁻/IL2rg⁻/⁻*) triple knock-out mice transplanted with human hepatocytes (*Bissig et al., 2010b*) (hFRG). The hFRG mice were generated commercially (Yecuris, Tualatin, OR) as described previously (*Bissig et al., 2010b*).

For data presented in *Figure 3B–D* and *Figure 3—figure supplements 1–2*, the study was carried out at the BSL2 facility of WuXi AppTec using approved IACUC protocols. Human hepatocyte-repopulated FRG mice with serum human albumin levels around 10.0 mg/mL were used for HBV challenge. The HBV virus (Genotype B, HBeAg negative) was obtained from an HBV-infected patient plasma sample with written consent. A total of 20 hFRG mice were challenged with high viral dose at $10^9$ GE HBV viruses in 200 μL per mouse by tail vein injection on day 0. All the animals were monitored on a daily basis for body weight changes and for clinical signs of viral infection for the duration of the in vivo study. For the 2H5-A14 prophylaxis group, five mice were injected with 2H5-A14 at 15 mg/kg by a single IP administration one day prior to HBV virus challenge. For the 2H5-A14 and HBIG treatment groups, the treatment started on day five post infection and lasted until day 21. 2H5-A14 or HBIG were administrated every three days by IP injection at 20 mg/kg and 40 IU/kg (72 mg/kg), respectively. For both the prophylaxis and the treatment models, blood samples were collected at different time points. These samples were used for quantifying viral HBsAg, HBV DNA, and RNA copies in sera. The mice were sacrificed at the end of the experiment at 37 dpi, and the liver tissues from all mice were harvested. One portion of the liver tissues was frozen in liquid nitrogen immediately upon harvest to be used for analysis of HBV DNA, RNA, and cccDNA. The remaining portion of the liver sample was fixed with 4% PFA in neutral PBS and paraffin-embedded for IHC analysis of HBsAg and HBcAg.

For comparing the effect of 2H5-A14 and 2H5-A14-DANA in HBV-infected hFRG mice (*Figure 5B–G*, and *Figure 5—figure supplement 1*), the study was carried out at NIBS's BSL2 animal facility according to approved IACUC protocols. The hFRG mice used this experiment were purchased from Yecuris. These mice had serum human albumin levels around 2.0–3.0 mg/mL. The HBV virus (Genotype D) was prepared from HepDE19 cells (*Sells et al., 1988*). A total of nine hFRG mice were challenged with high viral doses ($2 \times 10^9$) mge HBV per mouse by tail vein injection on day 0. All the mice were monitored on a daily basis for body weight changes and for clinical signs of viral infection for the duration of the in vivo study. PBS, 2H5-A14, or 2H5-A14-DANA was administrated by IP starting on day 33 post infection, twice a week until day 60, n = 3 for each group. Antibodies were given at 5 mg/kg. Blood samples were collected periodically until dpi 89 and used for quantifying viral HBsAg as described above and HBV DNA copies in sera. One mouse in the PBS control group was found dead with an undetermined cause at dpi 83 and excluded for blood collection and liver tissue collection at the end of the experiment (dpi 89). The rest eight mice were euthanized at dpi 89, and the liver tissues were harvested. One portion of the liver tissues was frozen in liquid nitrogen immediately upon harvest to be used for analysis of HBV viral DNA. The remaining portion of the liver sample was fixed with 4% PFA in neutral PBS and paraffin-embedded for IHC analysis of HBsAg.

## Quantification of HBsAg in serum samples of HBV-infected hFRG mice

HBsAg levels in mouse serum samples were measured using commercially available kits (Autobio Diagnostic CO). Serum samples were diluted 100 to 5000 times depending on HBsAg levels in the samples.

## Quantification of HBV DNA copies in serum samples

1 μL serum was diluted with 8.5 μL ddH$_2$O, to which was added 0.5 μL 1M NaOH. Samples were mixed well and incubated 100℃ for 10 min. Then, 2 μL was taken to use as a qPCR template. The qPCR was performed with a SYBR Premix Ex Taq kit (Takara) using HBV DNA specific primers: 5'-GAGTGTGGATTCGCACTCC-3' (forward) and 5'-GAGGCGAGGGAGTTCTTCT-3' (reverse) using an ABI 7500 Fast Real-Time system instrument (Applied Biosystems). The viral DNA copies were calculated based on a standard curve generated from samples with known copy numbers.

## Quantification of HBV RNA in serum samples

Total RNA were extracted from 1 μL of each serum sample diluted in 199 μL of 0.9% NaCl solution using a TIANamp Virus DNA/RNA Kit (Tiangen). The extracted RNA were treated with DNase I and then reverse-transcribed into cDNA with a PrimeScript RT kit (Takara) in a 20 μL reaction. 2 μL of cDNA was used as a template for qPCR as described previously(*Yan et al., 2012*). Primers

HBV1805F: 5'-TCACCAGCACCATGCAAC-3' and HBV1896R: 5'-AAGCCACCCAAGGCACAG-3' were for HBV-specific RNA.

## IHC assay

Chimeric mouse livers fixed with 4% PFA in neutral PBS and paraffin-embedded were processed for IHC analysis of HBsAg and HBcAg using a standard IHC protocol. For *Figure 3D* and *Figure 3—figure supplement 2*, intrahepatic HBsAg and HBcAg were detected using the specific Goat polyclonal anti-Hepatitis B Virus Surface Antigen (Abcam, UK), Rabbit polyclonal Anti-Hepatitis B Virus Core Antigen (Dako, Denmark), Donkey polyclonal Secondary Antibody to Goat IgG-H and L(AP) (Abcam), and Donkey polyclonal Secondary Antibody to Rabbit IgG-H and L(HRP) (Abcam), respectively. An AP-red substrate kit (Zhongshan Jinqiao Company, China) and ImmPACT DAB Peroxidase Substrate (Vector laboratories,Burlingame, CA) were used for staining. For *Figure 5E* and *Figure 5—figure supplement 1*, intrahepatic HBsAg were detected using a mouse mAb against Hepatitis B Virus Surface Antigen (clone: 56A1), followed by HRP conjugated anti-mouse IgG1 (Boster Biological Technology,Pleasanton, CA). DAB substrate (Brown) kit (Boster) was used for developing the HBsAg staining signals, nuclei were stained by Hematoxylin. Consecutive tissue sections were immunostained using a human-specific cytokeratin-18 (hCK18) mouse monoclonal antibody, clone DC10 (Dako) to visualize human hepatocytes. DAB substrate (blue violet) kit (Boster) was used for developing the hCK18 staining signals, nuclei were stained using Nuclear Fast Red (Boster).

## Southern blot analysis of total HBV DNA replication intermediates

Approximately 100 mg of frozen liver tissue were ground in a mortar and lysed in a lysis buffer (20 mM Tris, 0.4 M NaCl, 5 mM EDTA, 1% SDS, pH = 8.0) in the presence of proteinase K (QIAGEN) at 56°C overnight. After RNase A digestion, the cell lysate was extracted twice with phenol:chloroform: isoamyl alcohol (25:24:1, pH = 8.0). The extracted genomic DNA was precipitated with equal volumes of isopropanol at −20°C overnight. The DNA pellet was washed with 70% ethanol and dissolved in TE buffer (10 mM Tris-HCl, 1 mM EDTA, pH = 8.0), and digested with HindIII (NEB, Ipswich, MA) before being analyzed by Southern blotting. The DNA samples of normal mouse livers were prepared as well and used as negative controls in the Southern blotting. For *Figure 3—figure supplement 1*, about 500 ng of the prepared DNA samples were loaded one sample per lane and separated on 1.2% agarose gels and then transferred to a positively charged nylon membrane. The membranes were probed with DIG-labeled full-length HBV DNA to detect the HBV sequence, and membrane hybridization was detected using X-ray film. For *Figure 5F*, about 20 μg of the prepared DNA was separated on 1.2% agarose gel electrophoresis and transferred onto Amersham Hybond-N$^+$ membrane (GE Healthcare). The loading amount for each sample was normalized to the copy number of hNTCP DNA quantified by qPCR in the same DNA sample (see details in the method section of 'Quantification of HBV cccDNA in mouse liver tissues by qPCR'). For Southern blot, the Hybond-N$^+$ membrane was crosslinked and subsequently probed with $^{[\alpha\text{-}32P]}$dCTP (250 μCi, Perkin Elmer)-labeled HBV genotype D (Accession number: U95551.1) linear full-length genomic DNA. After overnight hybridization in PerfectHyb plus hybridization buffer (Sigma) at 65°C, the membrane was washed and exposed to Carestream X-OMAT BT Film (XBT-1). 100 pg each of 3.2 kb, 2.1 kb and 1.7 kb HBV DNA fragments prepared by PCR amplification of a plasmid containing 1.0 copies linear HBV genotype D genome was used as DNA marker.

## Southern blot analysis of HBV cccDNA

The cccDNA was selectively extracted using a protein-free Hirt method (*Hirt, 1967*) as previously described, with modifications. Briefly, to selectively extract HBV cccDNA, 100 mg of frozen liver tissue was lysed with 7.3 mL lysis buffer at room temperature for 30 mins, followed by addition of 2 mL of 2.5 M KCl and incubation at room temperature overnight. The lysis buffer was not supplemented with proteinase K, containing 1 mM Tris–HCl, pH 7.5, 10 mM EDTA, 150 mM NaCl, 10% SDS. The lysate was then clarified by centrifugation at 12,000 g for 30 min at 4°C and extracted with phenol and phenol:chloroform. DNA was precipitated with equal volume of isopropanol and finally dissolved in TE buffer (10 mM Tris, 1 mM EDTA, pH8.0). For Southern blotting, the DNA samples were separated on a 1.2% agarose gel and then transferred to a nylon membrane (Hybond-N+; GE-RPN2250B) using a standard neutral transfer procedure. 3.2 kb, 2.0 kb, and 1.3 kb HBV fragment

DNA was also run on the same agarose gel to serve as the molecular marker. The membrane was probed with the DIG-labeled HBV DNA probe. Hybridization was carried out in 10 ml of hybridization buffer with a 1 hr prehybridization at 60°C and overnight hybridization at 60°C, followed by 2 × 5 min wash with 2 × SSC, 0.1% SDS at room temperature and 2 × 5 min wash with 0.2 × SSC, 0.1% SDS at 60°C. The membrane was incubated with blocking buffer for 30 min, followed by 30 min incubation with antibody solution. After equilibration with detection buffer for 5 min, the membrane was rinsed with CDP-star and then exposed to X-ray film at room temperature.

## Quantification of HBV cccDNA in mouse liver tissues by qPCR

HBV cccDNA was quantified by qPCR using a protocol as previously described (*Qi et al., 2016*), with modifications. Briefly, the frozen liver tissues were homogenized and then lysed in a lysis buffer (20 mM Tris, 0.4 M NaCl, 5 mM EDTA, 1% SDS, pH = 8.0) in the presence of proteinase K (QIAGEN). The total DNA was extracted according to a standard phenol-chloroform extraction protocol. 5U (0.5 µL) plasmid-safe adenosine triphosphate (ATP)-dependent deoxyribonuclease DNase (PSAD) (Epicentre Technologies, Madison, WI) was used to digest 500 ng of total DNA in 20 µL reaction volume, and incubated at 37°C for 8 hr to remove the linear genomic and HBV replication intermediate DNAs. The reaction was then incubated at 70°C for 30 mins to inactivate the DNase. Then, 2 µL of the digested DNA for each sample was used as a qPCR template for quantification of intrahepatic HBV cccDNA copies. HBV cccDNA specific qPCR primers are: 5'-TGCACTTCGCTTCACCT-3' (forward) and 5'-AGGGGCATTTGGTGGTC-3' (reverse). The hNTCP gene copy numbers, which represent the amount of human hepatocytes in the liver tissues of chimeric mice, were used to normalize the relative cccDNA amount in the infected cells for each sample. For quantification of human NTCP gene copies, 50 ng of the undigested DNA samples were used. Human NTCP specific primers (annealing to the second intron of hNTCP gene) are: 5'-TCCAGGAGCCACTTTCACCATAA-3' (forward) and 5'-AGCAGGGACAAGTGTCAGAACAGA-3' (reverse). The amount of HBV cccDNA or hNTCP gene copies were calculated using the standard curves generated from the standard plasmids with known copy numbers.

## Northern blot analysis of HBV RNA

Total RNA was isolated from frozen liver tissues using TRIZOl reagent. About 2 µg of total RNA per sample was separated on a 1.2% formaldehyde-agarose gel and blotted onto positively charged nylon membrane. DIG-labeled probes were prepared with a DIG RNA Labeling Kit (Roche, Switzerland). After hybridization, the membrane was washed and exposed to X-ray film.

## Cloning and sequencing the L gene of HBV

Total genomic DNA was isolated from frozen homogenized liver tissues collected from each individual mouse in the 2H5-A14 treatment group and the control group. L gene was amplified from the extracted DNA by PCR using a set of specific primers. Two PCRs were performed for each sample. The PCR products were then cloned by TA cloning. For each TA cloning, clones were randomly picked for sequencing the L gene.

## Accession number

The coordinates and structural factors have been deposited into the Protein Data Bank with accession code 5YAX.

## Acknowledgements

We thank W Chen, Z Cao, Y Li and X Tian and all other members in the Sui and Li labs for their technical assistance and discussions. We also thank the NIBS Animal Facility for their help in handling and care of mice, the NIBS Biological Resource Centre for DNA sequencing, and the staff at the Shanghai Synchrotron Radiation Facility beamline BL17U for assistance in data collection of crystals. This work was supported by the Ministry of Science and Technology, China (973 Program: #2012CB837600 to JS and #2014CB849600 to WL); the Thousand Young Talents Plan (China) to JS; National Natural Science Foundation of China (NSFC81525018 to WL, 31325007 to KY), and

National Science and Technology Major Project (China) to WL (2013Z × 09509102). This work was also supported by Beijing Municipal Commission of Science and Technology.

## Additional information

### Competing interests

Dan Li: DL is a co-inventor of patent applications of antibodies reported in this study. Wenhui Li: WL is a co-inventor of patent applications of antibodies reported in this study. Jianhua Sui: JS is a co-inventor of patent applications of antibodies reported in this study. The other authors declare that no competing interests exist.

### Funding

| Funder | Grant reference number | Author |
| --- | --- | --- |
| Ministry of Science and Technology of the People's Republic of China | 2012CB837600 | Jianhua Sui |
| Beijing Council of Science and Technology | | Wenhui Li<br>Jianhua Sui |
| National Natural Science Foundation of China | NSFC81525018 | Wenhui Li |
| Ministry of Science and Technology of the People's Republic of China | 2014CB849600 | Wenhui Li |
| Major National Science and Technology Project | 2013ZX09509102 | Wenhui Li |

The funders had no role in study design, data collection and interpretation, or the decision to submit the work for publication.

### Author contributions

Dan Li, Investigation, Methodology, Acquisition of data, Analysis and interpretation of data, Help writing original draft; Wenhui He, Ximing Liu, Sanduo Zheng, Yonghe Qi, Investigation, Methodology, Acquisition of data Analysis and interpretation of data; Huiyu Li, Fengfeng Mao, Juan Liu, Lijing Pan, Methodology, Acquisition of data Analysis of data; Yinyan Sun, Resources, Methodology; Kaixin Du, Methodology, Analysis of data; Keqiong Ye, Funding acquisition, Writing—review and editing, Analysis and interpretation of data; Wenhui Li, Conceptualization, Resources, Supervision, Funding acquisition, Validation, Investigation, Methodology, Writing—review and editing, Analysis and interpretation of data final approval of the manuscript to be published; Jianhua Sui, Conceptualization, Resources, Supervision, Funding acquisition, Validation, Investigation, Methodology, Writing—original draft, Project administration, Writing—review and editing

### Author ORCIDs

Dan Li ⓘ https://orcid.org/0000-0002-3670-4056
Keqiong Ye ⓘ http://orcid.org/0000-0001-6169-7049
Wenhui Li ⓘ https://orcid.org/0000-0003-1305-7404
Jianhua Sui ⓘ https://orcid.org/0000-0002-1272-9662

### Ethics

Human subjects: Human PBMCs or serum samples used in this study were obtained previously and stored by the biologics research center at National Institute of Biological Sciences (NIBS). These anonymous samples were from donors with written informed consent. The use of these samples was approved by the IRB of NIBS (IRB reference numbers are IRBS03001 and IRBS030902), and followed the World Medical Association Declaration of Helsinki for medical research.

Animal experimentation: This study was performed in strict accordance with the recommendations in the Guide for the Care and Use of Laboratory Animals of the National Institutes of Biological Sciences (NIBS) and WuXi AppTec (Shanghai). All animal experiments were approved by the Institutional Animal Care and Use Committee (IACUC) of NIBS (protocol numbers M0028 and 0023), and by the IACUC of WuXi AppTec (protocol number #R20140319-Mouse-B).

## Decision letter and Author response

Decision letter https://doi.org/10.7554/eLife.26738.036
Author response https://doi.org/10.7554/eLife.26738.037

# Additional files

## Supplementary files
• Transparent reporting form
DOI: https://doi.org/10.7554/eLife.26738.030

## Major datasets
The following dataset was generated:

| Author(s) | Year | Dataset title | Dataset URL | Database, license, and accessibility information |
|---|---|---|---|---|
| Ximing Liu, Sanduo Zheng, Keqiong Ye, Jianhua Sui | 2017 | Crystal structure of a human neutralizing antibody bound to a HBV preS1 peptide | http://www.rcsb.org/pdb/explore/explore.do?structureId=5YAX | Publicly available at the RCSB Protein Data Bank (accession no. 5YAX) |

The following previously published dataset was used:

| Author(s) | Year | Dataset title | Dataset URL | Database, license, and accessibility information |
|---|---|---|---|---|
| Hayer J, Jadeau F, Deleage G, Kay A, Zoulim F, Combet C | 2013 | HBVdb: a knowledge database for Hepatitis B Virus | https://hbvdb.ibcp.fr/HBVdb/HBVdbDataset?view=/data/proteins/alignments/A_LHBs.clu&seqtype=2 | Publicly available at the Hepatitis B Virus database website (https://hbvdb.ibcp.fr/HBVdb/HBVdbIndex) |

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
