## [Decision Letter]

Thank you for submitting your article "A potent human neutralizing antibody Fc-dependently reduces established HBV infections" for consideration by *eLife*. Your article has been reviewed by three peer reviewers, one of whom is a member of our Board of Reviewing Editors, and the evaluation has been overseen by the Reviewing Editor and Tadatsugu Taniguchi as the Senior Editor.

The reviewers have discussed the reviews with one another and the Reviewing Editor has drafted this decision to help you prepare a revised submission.

Summary:

Based on the previous discovery that NTCP is a functional receptor for viral entry of HBV and HDV, this study aims to identify bnAb for HBV/HDV, and if any, to examine its potential usefulness for therapy against HBV/HDV. All of the reviewers recognize that the data are well controlled and presented, and that the overall technical quality is very high. The manuscript is well written and its impact is high. However, several concerns are raised by the three reviewers, and they discussed which point should be requested to make this manuscript stronger.

Essential revisions:

Since it has been thought that Abs are ineffective at terminating on ongoing infection, to decide the start timing of Ab treatment (Figure 3) following full establishment of chronic infection would further strengthen the impact as therapeutics. In this regard, instead of five days later in humanized chimeric mice, when Ab treatment is carried out later following full establishment of chronic infection, assessment of therapeutic effects and liver toxicity would be highly recommended. In regard to liver toxicity, at least human albumin and ALT levels should be examined longitudinally in addition to histopathlogy-ideally viral load.

---

## [Author Response]

Essential revisions:

*Since it has been thought that Abs are ineffective at terminating on ongoing infection, to decide the start timing of Ab treatment (Figure 3) following full establishment of chronic infection would further strengthen the impact as therapeutics. In this regard, instead of 5 days later in humanized chimeric mice, when Ab treatment is carried out later following full establishment of chronic infection, assessment of therapeutic effects and liver toxicity would be highly recommended. In regard to liver toxicity, at least human albumin and ALT levels should be examined longitudinally in addition to histopathlogy-ideally viral load.*

We agree that starting Ab treatment following full establishment of chronic infection would further strengthen the impact of nAb as therapeutics. We have conducted two mouse studies shown in Figure 3 and Figure 5, respectively. In the first mouse study, Ab treatment was initiated on day 5, post infection, while in the second study, the treatment was started 33 days after the HBV infection. As we discussed in the manuscript (Discussion section), the virus used in the first mouse study (Figure 3) had rampant viral propagation, which had adverse effect on the general health of these immune deficient hFRG mice, thus the study was ended at thirty seven days post infection. In the second animal study shown in Figure 5, viral propagation rate was slower than the first study, allowing us to assess the antiviral effect of 2H5-A14 over a longer period. The Ab treatment was started at 33 days after infection, when the infection was clearly established although it still kept going up. Under this condition, Ab treatment markedly reduced the level of infection, demonstrating that 2H5-A14 can reduce the extent of an established HBV infection.

To access liver toxicity, we measured the levels of human albumin and ALT in longitudinal serum samples of which sufficient amounts were available for the assays. For both albumin and ALT levels, there were no significant differences between 2H5-A14 treated group and other groups including vehicle control, 2H5-A14 prophylaxis group and HBIG control group. The result is presented as a new supplemental figure: Figure 3—figure supplement 1.